

# A lightweight and secure online/offline cross-domain authentication scheme for VANET systems in Industrial IoT

Haqi Khalid[1], Shaiful Jahari Hashim[1], Sharifah Mumtazah Syed Ahmad[1], Fazirulhisyam Hashim[1] and Muhammad Akmal Chaudhary[2]

[1] Department of Computer and Communication & Systems Engineering, Faculty of Engineering, Universiti Putra Malaysia, Serdang, Selangor, Malaysia
[2] Department of Electrical Engineering, College of Engineering, Ajman University, Ajman, United Arab Emirates

Corresponding authors
Haqi Khalid,
haqikhalid1@gmail.com
Shaiful Jahari Hashim,
sjh@upm.edu.my

## ABSTRACT

In heterogeneous wireless networks, the industrial Internet of Things (IIoT) is an essential contributor to increasing productivity and effectiveness. However, in various domains, such as industrial wireless scenarios, small cell domains, and vehicular *ad hoc* networks, an efficient and stable authentication algorithm is required (VANET). Specifically, IoT vehicles deal with vast amounts of data transmitted between VANET entities in different domains in such a large-scale environment. Also, crossing from one territory to another may have the connectivity services down for a while, leading to service interruption because it is pervasive in remote areas and places with multipath obstructions. Hence, it is vulnerable to specific attacks (*e.g.*, replay attacks, modification attacks, man-in-the-middle attacks, and insider attacks), making the system inefficient. Also, high processing data increases the computation and communication cost, leading to an increased workload in the system. Thus, to solve the above issues, we propose an online/offline lightweight authentication scheme for the VANET cross-domain system in IIoT to improve the security and efficiency of the VANET. The proposed scheme utilizes an efficient AES-RSA algorithm to achieve integrity and confidentiality of the message. The offline joining is added to avoid remote network intrusions and the risk of network service interruptions. The proposed work includes two different significant goals to achieve first, then secure message on which the data is transmitted and efficiency in a cryptographic manner. The Burrows Abdi Needham (BAN logic) logic is used to prove that this scheme is mutually authenticated. The system's security has been tested using the well-known AVISPA tool to evaluate and verify its security formally. The results show that the proposed scheme outperforms the ID-CPPA, AAAS, and HCDA schemes by 53%, 55%, and 47% respectively in terms of computation cost, and 65%, 83%, and 40% respectively in terms of communication cost.

## INTRODUCTION

The Industrial Internet of Things (IIoT), also known as the industrial Internet, put forward the IoT advances in development (*Shaikh, Zeadally & Exposito, 2015*; *Khalid et al., 2020a*). IIoT integrates a wide range of existing industrial automation systems with the latest electronics, computing, machine learning, and communication technologies. IIoT claims that in gathering and communicating data, intelligent machines are more capable than humans (*Khalid et al., 2021a*). This data makes business intelligence activities simpler for the manufacturing and business communities (*Sey, 2018*). An extensive network of vehicles and roadside units communicating with each other to share information is the *ad hoc* vehicle network, an IIoT application (*Latif et al., 2018*; *Al-Heety et al., 2020*). VANET is a particular case of wireless multihop network, which has the constraint of fast topology changes due to the high node mobility. With the increasing number of vehicles equipped with computing technologies and wireless communication devices, inter-vehicle communication is becoming a promising field of research, standardization, and development. VANETs enable a wide range of applications, such as prevention of collisions, safety, blind crossing, dynamic route scheduling, real-time traffic condition monitoring, etc. Another important application for VANETs is providing Internet connectivity to vehicular nodes (*Badis & Rachedi, 2015*). These are networks for naturally created needs from connected vehicles—VANETs aim to provide comfort for travelers and improve road safety and congestion. VANETs, information about vehicle-to-vehicle (V2V), and vehicle-to-infrastructure (V2I) communication between the highway and urban scenarios are shared wirelessly. The growing number of vehicles on the road causes many major traffic problems every day, including traffic delays and pileups of cars (*Kaiwartya et al., 2016*; *Khalid et al., 2017*). The industrial IoT is an emerging implementation of IoT technologies in several contexts, such as automation, intelligence controls, smart cities, smart transportation, and smart grids (*Rehman et al., 2021*).

It would be hard to incorporate industrial IoT solutions without the construction of an infrastructural network. It is important to understand unique IoT concepts when applying these methods to wireless IoT networks. One of the significant features of IoT networks is the collaboration between heterogeneous IoT devices. The Internet of Things (IoT) application areas have significantly increased as digital electronics and wireless networking evolve rapidly (*Goudarzi et al., 2019*). A broad range of technologies is currently funded, including industrial automation, smart transport, medical and e-health services (*Javed et al., 2020*). Low-weight, efficient communication between sensing devices and interoperability between various communication mechanisms is the IoT's critical issue (*Khalid et al., 2020b*). The industrial IoT data created from billions of device-person interactions will be massive and complex and will suffer from many security and privacy issues, particularly concerning device authentication. Computer security researchers have developed many authentication protocols, implemented in the industrial IoT context, to overcome these security concerns (*Ferrag et al., 2017*). Vehicle *ad-hoc* networks (VANETs), an essential part of an intelligent transport system, will use less wired communications technologies to provide continuing and reliable network

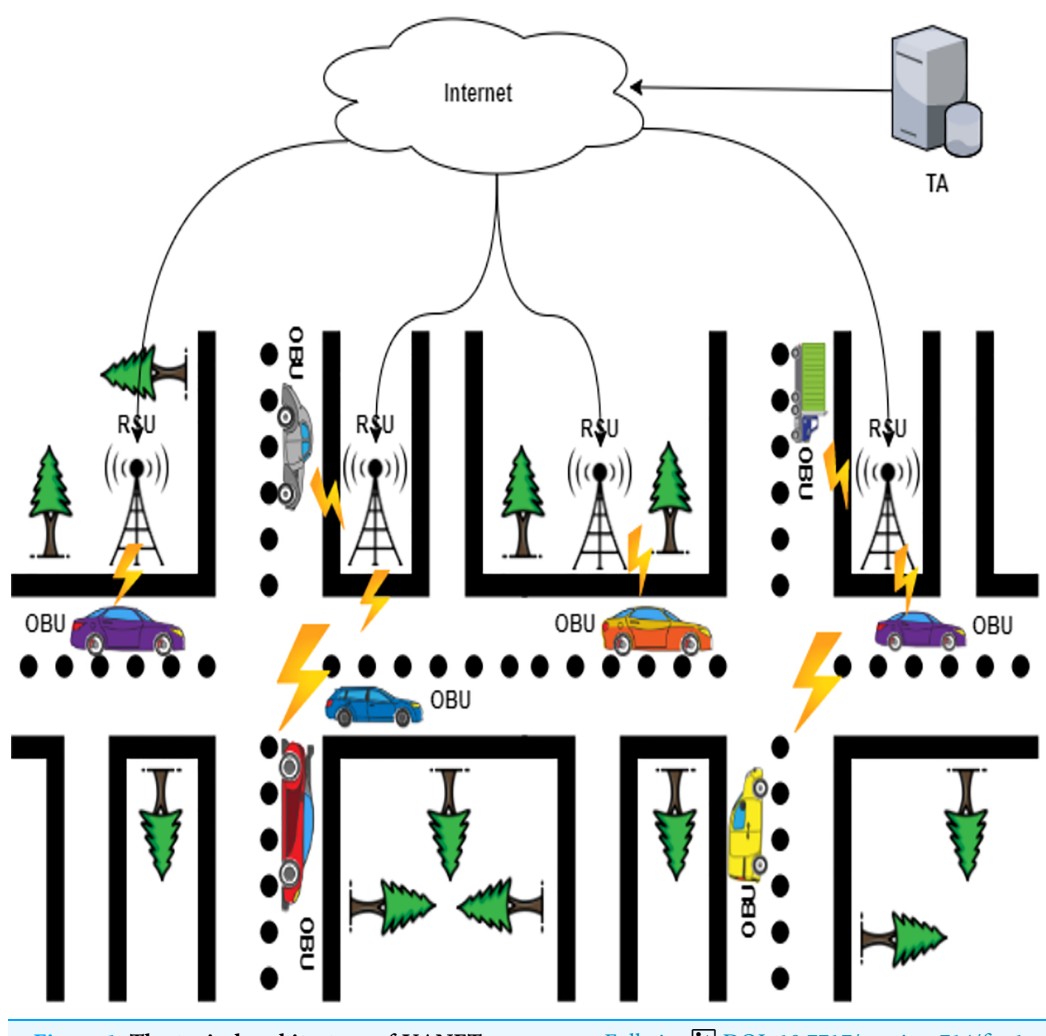

**Figure 1   The typical architecture of VANETs.**   

communications services (*Manvi & Tangade, 2017*). As illustrated in Fig. 1, VANETs are made of three essential entities: trust authority (TA), roadside units (RSU), and on-board vehicles (OBUs) (*Sheikh & Liang, 2019*).

- OBU: Each vehicle must be linked to the TA with the private key and the public device's necessary parameters. Secret information, such as private keys, is inserted into each vehicle's tamper-proof device to allow only authorized parties to access the tamper-proof device. Individual safe values, such as true vehicle identity and a secret vehicle key, are pre-loaded by the device. The vehicles' computation mechanism is also included in this system, and the hidden values are never revealed. OBUs routinely disseminate such data while traveling on roads, such as distance, current time, direction, speed, traffic conditions, and traffic events useful for other vehicles and RSUs. The 5.9 GHz Dedicated Short-Range Communication (DSRC) IEEE 802.11p is the communication protocol for neighboring OBUs.

- TA: TA has registered OBUs and RSUs. It initializes them with the public system's data or private keys. TA has a general computing and storage capacity and is the only party who can reveal the signers' identity. The solution to TA is impossible, and both parties to the scheme fully trust it.

- RSU: RSU is a stationary component system with DSRC wireless access point, stable memory storage, and computational capabilities. The time between requesting and receiving RSU responses is crucial for successfully disseminating data by VANETs, given the restricted transmission range of RSUs and vehicles' movement. RSUs are known as fully trusted parties in the scheme.

However, VANET architecture dealing with a hundred vehicle devices for accessing and management, this large amount of data and information seems to be a large-scale environment. However, these systems are limited resource devices in computation, storage, and energy. Traditionally, most authentication schemes rely on Roadside units (RSUs) that mainly hold the data's computing and processing. According to the large-scale architecture, the devices will deal with a large amount of data transmitted and processed. In a short time interval, several vehicles can continuously cross-practical areas of several RSUs. Also, at any time beyond prediction, the random vehicle can enter or leave the VANET network. The vehicles are also dynamically moved through different domains. This movement comes out with a critical problem across domain access. Because of the significant number of participating vehicles, the individual RSUs would have enormous time consumption and computation costs, which are crucial for limiting the comprehensive implementation of VANETs. Each vehicle and the RSU passed should be authenticated in time for each vehicle before exchanging vehicle data. Thus, this issue causes a significant delay and high computation costs, and it also increases the number of the interacted messages through a public network. Therefore, the VANET system will take a lengthy verification process before granting access (*Picone et al., 2015*). Likewise, transmitted data between the RSUs, OBUs, and the trusted party are sensitive. The adversaries are mainly targeting this information to delete, manipulate, eavesdrop on this data. Current authentication schemes are vulnerable to specific attacks (*e.g.*, replay attacks, modification attacks, man-in-the-middle attacks, and insider attacks), and these attacks make the VANET system week (*Deepa et al., 2021*). For example, a MiTM attack occurs in the middle of V2V communication to check closely and alter the messages. The attacker can access and control the entire V2V communication, but the communication entities think they can communicate directly in private. Also, this way, each vehicle's temporary identity changes over time, and a malicious attacker can hardly trace a specific vehicle. This is because after altering the certificate, an attacker would not link the new certificate with the old certificate, which means that the attacker has lost the target. However, this method still has some problems, such as high revocation costs. For example, when a vehicle is revoked, the number of pseudonymous certificates that need to be added to the Certificate Revocation List (CRL) could be too large. The size of CRL increases rapidly when the size of the network increases. These attacks could enable adversaries to enter the VANET system user's registered ID, password and broadcasting a false message, or

repeat/delay the transmission fraudulently (*Khalid et al., 2021b*). Also, preserving data confidentiality, privacy, and integrity in the trusted information context, where the information is shared between many parties, is becoming one of the most challenging issues for such a community. Therefore, a lightweight cross-domain authentication scheme for the VANET system is critically needed to satisfy the VANET's security requirements.

Motivated by the above discussion on VANET secure transmission, we proposed an online/offline lightweight authentication scheme for VANET in industrial IoT. The offline joining and handover phase was added to avoid service interruption if the connectivity is down, allowing vehicles to send authentication requests. At the same time, they are temporarily disconnected from the Internet (*Deepa et al., 2020*). In offline authentication, TA is not involved in the joining procedure since the information is preloaded prior. The combination comprises the Advanced Encryption Algorithm (AES) and the Database Encryption RSA algorithm for the integrity, authentication, and distribution of the key. The algorithms have less encryption and decryption time in processing such extensive data. This mechanism also provides dual protection by taking advantage of the algorithms used, so the data transmission in the network will be more secure. The main advantage of this combination is that the AES-RSA encryption algorithm utilized the features of already existing algorithms which are very secure and difficult to break since it requires two different keys and algorithms. The strength of the security is improved by combining symmetric and asymmetric encryption methods where retrieval of the key is very difficult. The scheme ensures resistance against specific attacks, *e.g.*, such as reply, modification, impersonation, and man-in-the-middle attacks. Also, it provides message integrity, authentication, and identity privacy preserving against change.

The study lists the findings as follows: in "VANETs security requirements," we identify the security requirements of the VANET system; in "Related Work," we review the previous studies and categorize their limitations; in "Preliminaries," we give a brief introduction on RSA, and AES-RSA algorithms; in "proposed Scheme" presents the main finding of the study; in "Security analysis" verify the security aspects using BAN logic, and AVISPA tool; in "performance evaluation" we evaluate the performance of the proposed scheme; in "Conclusion" the study is finalized, and a brief conclusion is given.

## VANETS SECURITY REQUIREMENTS

Vehicles in VANETs may detect nearby traffic details or an event to notify neighboring vehicles or the central traffic center. The authentication of messages can reduce these threats because of users' wrong behaviors, such as false information transmissions, re-transmission of previous messages, and changes in the messages sent. Since users' data should be kept secret, including driver names, speeds, positions, and relationships with other users, authentication should be performed anonymously (*Khan et al., 2021*). There is a contradiction between anonymity and dedication. As a result of anonymous authentication, unauthorized users should not utilize the network against external attackers (*Hemalatha, 2021*). If approved users do something wrong, anonymous authentication will not track them. For TA to determine the sender's real identity, anonymous authentication should therefore be performed. We thus need the preservation

of authentication protocols on conditional privacy. The security criteria for the VANETs are as follows:

1. **Message integrity and authentication:** VANETs must be sure to create and send the received message through an approved OBU and that nobody modifies the received message. Moreover, the authentication scheme should be immune to impersonation, and no signature vehicle could be impersonated (*Kumar & Singh, 2021*).

2. **Identity privacy-preserving:** The security of identity information underlines that by monitoring communications in VANETs, an intruder cannot identify either the initiators of the message or the party, including its originators. As vehicle names and locations are private and privacy disclosure is immoral, this is a critical property for VANETs.

3. **Traceability:** This means that TA can identify the identity of the originators if appropriate. VANETs are susceptible to insiders without traceability, and a malicious user can easily give the other vehicle a wrong message and fool them.

4. **Unlinkability:** Except for DTA, the RSU and the malicious vehicle should not determine two communications from the same vehicle.

5. **Resistance to attacks:** Various common attacks occur in VANETs, such as the impersonation attack, the alteration attack, the replay attack, the man-in-the-middle attack, and the stolen verifier table attack, should be able to withstand the system.

## RELATED WORKS

In recent years, security authentication and privacy protection have been a significant research orientation in VANETs. Several anonymous authentication schemes were suggested for VANETs. *Azees, Vijayakumar & Deboarh (2017)* proposed in 2017 an effective anonymous authentication scheme (EAAP) for VANETs. No storage of anonymous vehicle certificates and RSUs based on bilinear pairing is required by the trusted authority (TA) in the EAAP. In the case of a dispute, the trust authority will revoke and expose its real identity to a misbehaving vehicle's privacy. The revoked identity is then put on the TA's retained identity revocation list (IRL). Furthermore, without incentives, the enthusiasm problem still suffers when sending messages. *Verma et al. (2021)* presents a short digital signature scheme without pairing in a certificate-based setting with aggregation in an IIoT environment. In the SCBS scheme, each signer/user generates his/her (public and secret) keys and gets a certificate on (ID, public access) pair from CA. Certificates are sent *via* a public channel. During the execution of the signing phase, the signer requires his/her updated certificate along with a secret key. Similarly, *Moni & Manivannan (2021)* proposed a distributed and scalable privacy-preserving authentication and message dissemination scheme. Traditionally Certificates and CRLs were used for authenticating entities. However, as the number of entities grows, using CRLs for authentication incurs significant computation and communication overhead. In this scheme, a vehicle only needs to store the public key of the TA and the latest MHT root generation timestamp to authenticate RSUs. Similarly, MMPT is used by RSUs to

authenticate vehicles, thus reducing the complexity involved in authenticating vehicles. *Xie et al. (2017)* subsequently introduced a new, efficient authentication process, using identity to relatively protect VANET applicants' privacy. The ECC is used to solve the problem of the bilinear pairing because of its complex operations. The proposed system is an improved CPA solution based on (*He et al., 2015*) that is more effective than the former and fulfills VANET security requirements. The proposed scheme offers a simple message verification and batch message verification, where several messages can simultaneously be verified, and authentication costs are significantly reduced. However, a TA can track this vehicle when a vehicle broadcasts false information without preventing it from transmitting these messages. Furthermore, the identity of each vehicle can be easily discovered by an insider attacker since this attacker has private and public key pairs and has high computational and communication costs.

In *Vijayakumar et al. (2018)*, a signature-based anonymity technique was suggested for vehicular *ad hoc* networks using bilinear pairing. However, this method eventually introduces enormous computational complexity and overhead, which are unfeasible for the RFID Tag resource restriction. A conditional monitoring mechanism is developed through which the TA tracks the wrong vehicles or RSUs in the IoT environment that misuse the VANET. The TA will, therefore, revoke the privacy of misbehaved vehicles for additional damage. Efficient authentication of the anonymous batch message (ABM) also suggested testing the authenticity of an RSU while sending a batch of messages *via* RSU to vehicles. However, because of the high overhead of communication, the high computational cost of the Certificate Revocation List (CRL) testing method makes it difficult to validate a large number of VANET messages over a specific period (*Lu, Qu & Liu, 2018*). Similarly, *Pournaghi et al. (2018)*, proposed the NECPPA scheme, incorporating schemes based on RSU and TPD. The key concept for this system is that the master and public parameter is stored on the RSU TPD. This is because the connection between TA and RSU is secure and fast for communication. The RSU, therefore, generates the sub-master key inside the coverage area to be sent to all vehicles (*Zmezm et al., 2015*). The execution time during message generation and verification, however, is high (*Al-Shareeda et al., 2020*). *Li et al. (2018)* a conditional anonymous authentication of the VSNs' privacy was proposed, while the authors suggested the VSNs' design goals. Their scheme is robust and adopts pseudo-identity generation and private key extraction to maintain anonymity. To keep the privacy of its identity, every OBU should restore several pseudo-identities in this scheme. This scheme promotes the security and privacy needed for services rendered by VANET. However, the machine's private key is pre-loaded into the car's tamper-proof computer, which attackers can eliminate (*e.g.*, through side-channel attacks). Hence, when the attackers have physical access to the tamper-proof device, their solution is not secure.

Likewise, an available certificates conditional privacy-preserving authentication scheme for vehicular ad-hoc networks was proposed by *Ming & Shen (2018)*. Certificateless cryptography and elliptical curve cryptography form the basis of the proposed scheme (ECC). As an adversary would not connect a vehicle to its transmitted message, the system encourages conditional privacy and ensures unlinkability. In this work, however, the

property of non-observability was not considered. *Zhong et al. (2019)* proposed a privacy protection scheme for safe V2I communications based on a certificateless aggregate signature, and the scheme could achieve complete aggregation. It utilizes the RSU as the aggregator to aggregate under its coverage the signatures signed by the vehicle. The authors attempted to fix the problem in the verification step and had a significant overhead in the signature authentication process. Unfortunately, their latest scheme uses the bilinear pairing operation and the Map-To-Point hash function in the verification process, which has added high overhead in verifier computation expense. A message verification scheme has been suggested for VANET (*Cui et al., 2018*). However, it is still not comparatively efficient due to the need for many EC operations, and the overhead for communication is high. The system (*Cui et al., 2018*) is vulnerable to attacks by impersonation, alteration, man-in-the-middle, and concatenation. A protocol for the vehicular environment was also proposed in 2018 by *Mukherjee, Gupta & Biswas (2019)*. In this scheme, lattice-based cryptography is used. This scheme is secure in a quantum computing system, but the identity and password are stored directly in a tamper-proof device. If an opponent catches a TRD, then details may be leaked *via* the side-channel attack. A mutual authentication scheme was subsequently proposed for V2V in the *ad hoc* vehicle network to achieve better efficiency and security (*Xie et al., 2017*). Using elliptic curve encryption technology, the authors attempted to perform privacy-preserving mutual authentication for regular V2V communication. Sadly, their method is vulnerable to man-in-the-middle attacks and modification attacks. In 2020, instead of a map-to-point hash for safe V2I communication, *Ali & Li (2020)*, using BP and a general one-way hash, introduced an ID-based framework. The messages are authenticated easily by an RSU within their scheme. Instead of map-to-point hash functions, it utilizes general one-way hash functions during high traffic density area verification. Since the private key generator (PKG) has access to all users' private keys in identity-based schemes, the main escrow problems will occur if PKG was compromised. Lightweight security was suggested without using a single verification batch verification system (LSWBVM) scheme to broadcast many safety messages while driving (*Al-Shareeda et al., 2020*). However, because the verifying vehicle for signature authentication uses only a one-way hash feature, this system is vulnerable to various security threats, such as impersonation and alteration attacks. Also, since the timestamp is not included in the safety message tuple, it is prone to replay attacks. Besides the authentication and honesty requirements, this scheme does not meet in-vehicle systems. Moreover, since the name of the vehicle stored on TPD has not been updated for a long time, it is suspected of side-channel attacks.

In 2020, an anonymous authentication scheme based on community signature in VANETs was proposed by *Jiang, Ge & Shen (2020)* (AAAS). As a group manager, AAAS adds a regional trust authority (RTA) to provide anonymous vehicle authentication and communication services that can efficiently increase TA's computing and communication costs and alleviate RSU pressure with low computing and storage capacity. However, the high traffic congestion increases the number of messages transmitted, which increases the overhead of computations and communications from VANET. A refiling framework has been developed for on-demand pseudonyms and certificates by

*Benarous et al. (2020)*; anonymous tickets and challenge-based authentication are the foundation of their scheme. The scheme's effectiveness against the most popular security parameters is tested using several methods and techniques that have proven its efficiency and robustness, such as the BAN logic, SPAN, and AVISPA instruments. Recently, *Alfadhli et al. (2020b)* proposed a novel and successful CPPA-VANET solution based on lightweight pseudo-identity to overcome the crucial driving area and key escrow problems and provide better efficiency in terms of computation cost and overhead communications. Regrettably, the device also has a high computational cost in the authentication process and is prone to replay attacks. Similarly, *Cheng & Liu (2020)* an improved ECC authentication scheme based on RSU was proposed, in which RSU distributes vehicle pseudonyms when the vehicle pseudonyms are invalid. However, the password is estimated to have a low entropy secret value and vulnerable to password guessing attacks due to the built-in issues related to the password.

In *Thumbur et al. (2020)*, to avoid the complicated public fundamental infrastructure certificate management problem and the Identity-based key escrow problem, a new VANET certificateless aggregate signature-based authentication scheme was proposed. All signatures/messages received from the surrounding vehicles are aggregated into a single signature by the RSU. AS/RSU can ensure that the related messages are signed by only the registered vehicles. The lack of an effective signature authentication process, however, increases the overhead of computing. *Jiang, Ge & Shen (2020)* and *Jiang, Hua & Wahab (2020)* also proposed a Self-checking Authentication Scheme with Higher Efficiency and Security for VANET, called SAES; the proposed scheme adopts pseudonym-based self-checking authentication. Unfortunately, the system also suffers from primary session attacks, modification attacks, and high processing costs due to the bilinear pairing. Similarly, for VANETs that protect privacy, a lightweight multi-factor authentication and security solution was introduced (*Alfadhli et al., 2020a*). It operates as authentication variables, a mixture of physically unclonable (PUF) functions and one-time dynamic pseudo-identities. The proposed scheme removes the need for a TPD to store sensitive long-term data (such as a fingerprint, password), enhancing the system's effectiveness and security. Nevertheless, by analyzing the content of such captured messages in VANETs, an intruder can acquire the original identity and track its traveling routes. From the above analysis, we found out that most of the existing schemes suffer from high computation and communication costs because the architecture of VANET contains a considerable quantity of vehicles. Likewise, transmitted data between the RSUs, OBUs, and the trusted party are sensitive. The adversaries are primarily targeting this information to delete, manipulate, eavesdrop on this data. Some attacks (*e.g.*, replay attacks, modification attacks, man-in-the-middle attacks, and insider attacks) are vulnerable to current authentication systems, and these attacks make the VANET system week. Such attacks will probably allow adversaries to access the registered ID of the VANET device user and password and broadcast a false message or fraudulently repeat/ delay the transmission. Though some research attention has been paid to date, the critical issue of cross-domain authentication has not been appropriately addressed in the VANET market. As a matter of fact, under the static trust model, most of the existing VANET

authentication mechanisms tend to build up the verification process, where only the initial RSU opportunity is discussed. The CDA ability, in other words, was not considered at all. Both successive RSUs must request sensitive information from the cloud server for the remaining systems where the CDA issue has already been solved, causing unnecessary contact burdens and high latency. The comparison of the existing studies is shown in Table 1.

## PRELIMINARIES

In this section, the mathematical concept of RSA and the AES-RSA algorithm steps proposed are discussed. First, the basic definition and properties of the RSA algorithm are highlighted to explain RSA encryption and decryption. The combined AES-RSA algorithm is also described to understand the workflow on the sender and receivers' sides. Figure 2 shows the workflow diagram of the AES-RSA algorithm.

### RSA cryptosystem

Here, the basic description of the RSA cryptosystem and its properties are discussed. Two appropriate primes $p$, $q$ and $n = p * q$ are selected by Server TA as well as $(n) = (p - 1) * (q - 1)$. TA is now choosing an integer $e$ such that $gcd(e, (n)) = 1$. Further, TA computes $de - 1\ mod(n)$. Finally, the public key for TA is $(e, n)$, and $d$ is the private key. The algorithm's description is given as:

- Encryption: OBUs take the message $m$ and the public key $e$ from TA in RSA encryption and encrypt the message as $c = m^e$ and send the output c to TA.
- Decryption: TA takes cipher c and its private key d on the RSA decryption server and decrypts cipher c as $m = c^d$ and gets the message.

### AES-RSA encryption/decryption

The AES-RSA algorithms' steps on both sides, sender, and receiver are shown in this section. The steps are shown as follows:

**Encryption:**

1. User data, *i.e.*, identity and information, are given input to the AES and SHA-2 algorithms.
2. SHA-2 is hashing algorithm used to generate the hash value of the given plaintext.
3. The RSA is used to encrypt the hash value using the public key and produce the digital signature.
4. The plaintext is also encrypted with an AES using the AES's public key.
5. Then, the RSA public key is used to encrypt the text encrypted with an AES.
6. The digital signature is now padded with an AES encrypted text and sent through the cross-domain Internet to the receiver side.

**Table 1** Comparison of the existing authentication schemes in VANET.

| References | Issue | Structure | Method | Tool | Objective | Evaluation Parameters | Limitation |
|---|---|---|---|---|---|---|---|
| Azees, Vijayakumar & Deboarh (2017) | Malicious vehicle entering in the VANETs | Centralized | Bilinear pairing | Cygwin 1.7.35-15, PBC library | Track the vehicles that misuse the VANET or road-side units | Computational cost and signature verification process | Suffers from the problem of enthusiasm when forwarding messages |
| Verma et al. (2021) | 2021 Security issues, such as authentication, integrity, and confidentiality | Centralized | ECC | JCA library and JPBC library | Removes the certificate revocation queries in PKC | Computation cost, Communication cost | Vulnerability to attacks (e.g., insider attack, server spoofing attacks) |
| Moni & Manivannan (2021) | Significant computation and communication overhead | Centralized | Merkle HashTree | Crypto++ | Reducing the complexity involved in authenticating vehicles | Computation cost, Communication cost | Key session attacks and replay attacks vulnerability |
| Xie et al. (2017) | OBUs and RSUs are constrained in computing and cannot afford the verification of large messages | Centralized | ECC | MIRACL library | Ensures security and integrity for V2V and V2I communication messages | Computation cost, Communication cost | Any vehicle's real identity can be easily discovered by sufferers of high computing and communication costs and an insider attacker |
| Vijayakumar et al. (2018) | High computational cost in the process of checking the certificate revocation list (CRL) | Centralized | Bilinear pairing | PBC library | Provide a conditional tracking framework in which the TA traces the misbehaving vehicles or RSUs | Computational cost | Suffers high communication overhead |
| Pournaghi et al. (2018) | Increasing the number of revoked users allows the CRL volume to increase dramatically, which increases the signature verification period | Centralized | ECC | OMNET ++ | Provide a secure and fast communicational link between TA and RSU | Computation cost, communication cost | The execution time during message generation and verification are high |
| Li et al. (2018) | Elevated computing criteria during certificate generation and message verification phases | Centralized | ECC, pseudo-identity | PBC library | To improve efficiency further | Computation and communication overheads | If attackers have physical access to the tamper-proof device, it is not secure |
| Ming & Shen (2018) | Wrong output due to map-to-point hash and bilinear pairing operations requirements | Centralized | Certificateless cryptography and ECC | MIRACL Crypto SDK, ns-3.26 simulator | Reduce the cost of computing and communication | Computation and communication costs | Vulnerability to attacks (e.g., insider attack, server spoofing attacks) |
| Zhong et al. (2019) | Large overhead in the signature authentication process | Centralized | Certificateless aggregate signature | MIRACL library | Reduce the computation cost in the sign phase | Computation and communication cost | Large overhead in the verification phase |

(Continued)

| References | Issue | Structure | Method | Tool | Objective | Evaluation Parameters | Limitation |
|---|---|---|---|---|---|---|---|
| Mukherjee, Gupta & Biswas (2019) | An adversary can easily track a mobile node's route and the privacy of its driver | Centralized | lattice-based cryptography | PBC library | Assure secure communication | Computation and communication costs | Side-channel attack information could be leaked |
| Wu et al. (2019) | High computational complexity | Centralized | ECC | MIRACL library | Achieve better performance and security | Computation and communication costs | Vulnerable to man-in-the-middle attack and modification attacks |
| Ali & Li (2020) | Not successful in signing and checking a single message because of the comprehensive operations | Centralized | Bilinear pairing | JPBC library | Increases the efficiency | Computation and communication costs | Key escrow issues |
| Al-Shareeda et al. (2020) | Massive overheads in computation, especially in the batch verification phase | Centralized | ECC | MIRACL library | To verify many messages | Computation and communication overheads | Vulnerable to replay attacks |
| Jiang, Ge & Shen (2020) | The vehicle could not check the legal existence of the RSU response | Centralized | Pseudonym mechanism and group signature | JPBC library | To balance security and efficiency | Communication overhead, computation cost, and signaling cost | Increases the computations and communications overheads |
| Benarous et al. (2020) | To acquire pseudonyms, pseudonym refilling is still preferred | Centralized | ECC | PBC library | Ensure the user's unlinkability and anonymity | Computation and communication costs | High computation cost |
| Alfadhli et al. (2020b) | overcome the system key escrow problems | Centralized | Hash function only | PBC library | To protect the vehicle's privacy | Computation and communication costs | Key session attacks and replay attacks vulnerability |
| Cheng & Liu (2020) | Vulnerable to impersonation attacks and reveal the privacy of users during the communication process | Centralized | ECC | PBC library | Avoiding the risk of compromising the TPD of one vehicle leading | Computational and communication overhead | Password guessing attack |
| Thumbur et al. (2020) | The complex certificate management problem | Centralized | ECC | MIRACL library | Avoid key escrow problem | Computational and communication overhead | Signature checking increases the computation overhead |
| Jiang, Hua & Wahab (2020) | The batch verification can fail due to an invalid request problem | Centralized | pseudonym | PBC library, NS2.34 | Minimize the authentication cost | Computational, communication cost, average delay, and the packet loss ratio | High computation cost due to the utilized bilinear pairing |
| Alfadhli et al. (2020a) | Cloning or physical attack | Centralized | bilinear pairing | PBC library | Enhances the system security and privacy | Computational and communication overhead | Large overhead in the verification phase |

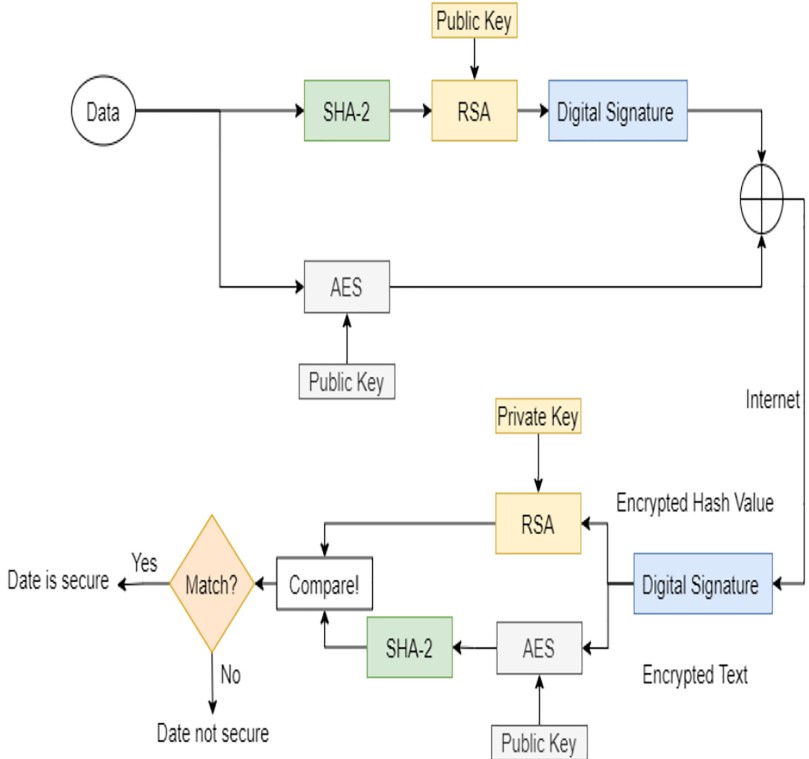

**Figure 2** **The AES-RSA algorithm work diagram.**

**Decryption:**

1. The receiver now receives the message it decrypts the digital signature using the sender's public key to retrieve the encrypted text and the hash value.
2. The retrieved encrypted text is decrypted using it is the public key to obtain the plaintext.
3. Then, the hashed value is decrypted into a message digest using the RSA's private key.
4. The decrypted text from the AES is passed to SHA-2, and the hash value is generated for the input plaintext.
5. The generated hash value is then compared to the one generated from the RSA and SHA-2 to check the message's validity.
6. If both are matched, then the integrity of the message is achieved.

## PROPOSED SCHEME

The lightweight authentication scheme for the VANET cross-domain system in industrial IoT is proposed in this section. The system includes entities such as the Trusted Authority (TA), the Domain Trusted Authority (DTA), road-side units (RSUs), and vehicles (Vi). The proposed scheme comprises eight phases: the setup phase, the vehicle registration phase, the domain TA registration phase, the RSU registration phase, the online joining phase, the online crossover phase, the offline joining phase, and the offline crossover

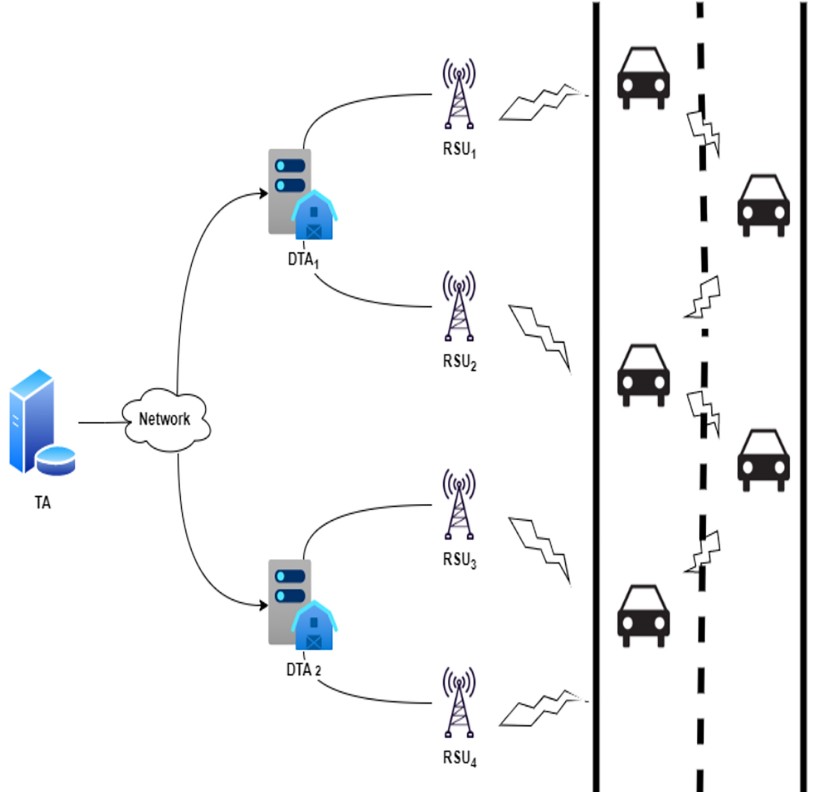

**Figure 3 Network diagram of the proposed scheme.**

phase. Figure 3 displays the proposed scheme's network diagram. The notations and definitions used in the scheme are shown in Table 2. The phases of the scheme proposed are described in detail below.

### Setup phase

To initialize the system, the trusted authority TA selects two large primes $p, q$ and computes $n = p, q$. The trusted authority TA keeps $p, q$ as secret parameters and publishes $n$ as a public parameter. Then, the trusted authority TA chooses a prime $e$ (*where* $1 < e < (p − 1)(q − 1)$) and computes d such that $ed1\ mod\ (p − 1)(q − 1)$. The trusted authority TA also chooses a one-way hash function $h(): 0, 1^* \to Z^* q$. The trusted authority TA publishes e as public and keeps d as secret. Also, the TA choose an encryption/decryption pair $Enc\{.\}$, $Dec\{.\}$ related to AES-RSA algorithm. The exchanged messages are encrypted using AES public key for secure transmission. The RSA public key is also used to encrypt the generated signature to provide integrity, confidentiality, and authenticity.

### Vehicle registration phase

In this phase, the vehicle must be registered at the trusted authority TA to authenticate to the distributed domains. The vehicle initializes the session by sending the identity and other security parameters to the TA *via* a secure channel. The transmitted message is protected where the information is double encrypted using the AES-RSA algorithm.

**Table 2 Notations.**

| Notation | Definition |
|---|---|
| TA | Trusted authority |
| DTA | Domain trusted authority |
| RSU | Road-side unit |
| Vi | Vehicle |
| $p, q$ | Large prime numbers |
| $h(\cdot) : \{0, 1\}$ | One-way hash function |
| $s \in Z_q^*,$ | TA's secret key |
| $VID_i$ | Vehicle's identity |
| $TA_{rsa}^{pk}$ | TA's RSA public key |
| $TA_{aes}^{pk}$ | TA's AES public key |
| $TA_{rsa}^e$ | TA's RSA private key |
| $t_{exp}$ | Expiration of secret key |
| $K_{TA \to v}, K_{v \to TA}$ | A key session between Vi and TA |
| $ID_{dta}$ | DTA identity |
| $K_{TA \to DTA},$ | A key session between TA and DTA |
| $ID_{rsu}$ | RSU identity |
| $K_{DTA \to RSU}$ | The key session between DTA and RSU |
| $r_v^j, r_2^{dta}, r_{rsu}$ | Random numbers |
| $Sign_{dta}$ | DTA signature |
| $Sign_{rsu_1}$ | RSU signature |
| $T_1, T_2, T_3$ | Timestamps |

When the TA receives the message, it checks the existence of the information in the database; if the vehicle is registered, the server will send a notification; otherwise, the vehicle performs the following steps as shown in Fig. 4.

1. Firstly, the Vehicle $V_i$ randomly picks a secret key $s \in Z_q^*$, secret value $R_i$, and computes $A_i = a.p$. Then, it computes $T_i = H(VID_i \parallel s)$, and encrypt the hash value with RSA's public key $Enc\_TA_{rsa}^{pk}\{T_i\}$. The vehicle parameters and it is identity are concatenated and encrypted with AES's public key $CT_{V \to TA} = Enc\_TA_{aes}^{pk}\{A_i, R_i, Enc\_TA_{rsa}^e\{T_i\}\}$. The vehicle sends the $CT_{V \to TA}$ to the TA.

2. The trusted authority TA receives the message $CT_{V \to TA}$ from the vehicle, it will decrypt the $CT_{V \to TA}$ using it is public-key $Dec\_TA_{aes}^{pk}\{A_i, R_i, Enc\_TA_{rsa}^{pk}\{T_i\}\}$ to obtain the encrypted identity and the parameters $< A_i, R_i, Enc\_TA_{rsa}^e\{T_i\} >$.

3. Then, it uses the RSA private key $Dec\_TA_{rsa}^d\{T_i\}$ to obtain the vehicle identity $VID_i$. TA will select a few random values $r_v^j \in Z_q^*$ to calculate vehicles pseudonyms $FID_v = H_3(VID_i, r_v^j)$ and corresponding public key $PK_v^j = H_1(ps_v \parallel t_e xp^v)$, and private keys $SK_v^j = d.PK_v^j$, where $t_e xp$ is the expiration of $r_v^j$, $1 < j < n, n$ is the total number of each vehicle obtaining pseudonym. Later, TA calculates the session key with the vehicle $K_{TA \to v} = d.A_i$ and encrypts $< r_v^j, SK_v^j, t_{exp}^v, R_i >$ to get

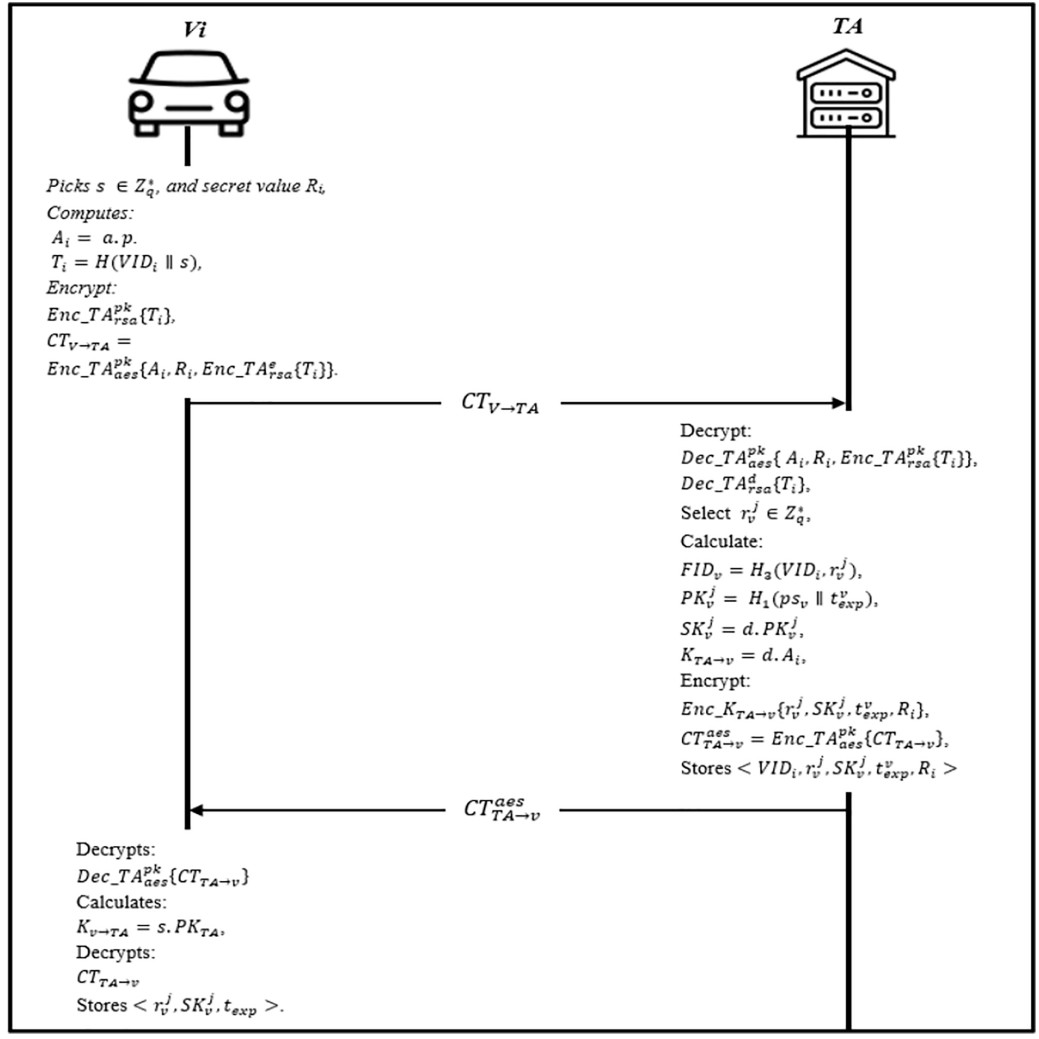

**Figure 4** Vehicle registration phase.

$CT_{TA \to v} = Enc\_K_{TA \to v}\{r_v^j, SK_v^j, t_{exp}^v, R_i\}$. Finally, it stores $< VID_i, r_v^j, SK_v^j, t_{exp}^v, R_i >$, and encrypt the ciphertext with AES public key $CT_{TA \to v}^{aes} = Enc\_TA_{aes}^{pk}\{CT_{TA \to v}\}$ and sends $CT_{TA \to v}^{aes}$ to the vehicle.

4. Upon receiving $CT_{TA \to v}^{aes}$ from $TA$, $Vi$ decrypts it $Dec\_TA_{aes}^{pk}\{CT_{TA \to v}\}$ to obtain $Enc\_TA_{aes}^{pk}\{CT_{TA \to v}\}$ and calculates the session with TA $K_{v \to TA} = s.PK_{TA}$ and decrypts $CT_{TA \to v}$ to obtain $< r_v^j, SK_v^j, t_{exp}, R_i >$. After obtaining $N_i$, vehicle verifies it and stores $< r_v^j, SK_v^j, t_{exp} >$. Otherwise, the vehicle needs to reapply for registration.

## Domain TA registration phase

This phase enables the domain trusted authority DTA to register itself into the trusted authority TA. The DTA sends a registration request containing the hashed value of the domain along with a freshly generated random number. Figure 5 shows the steps of the

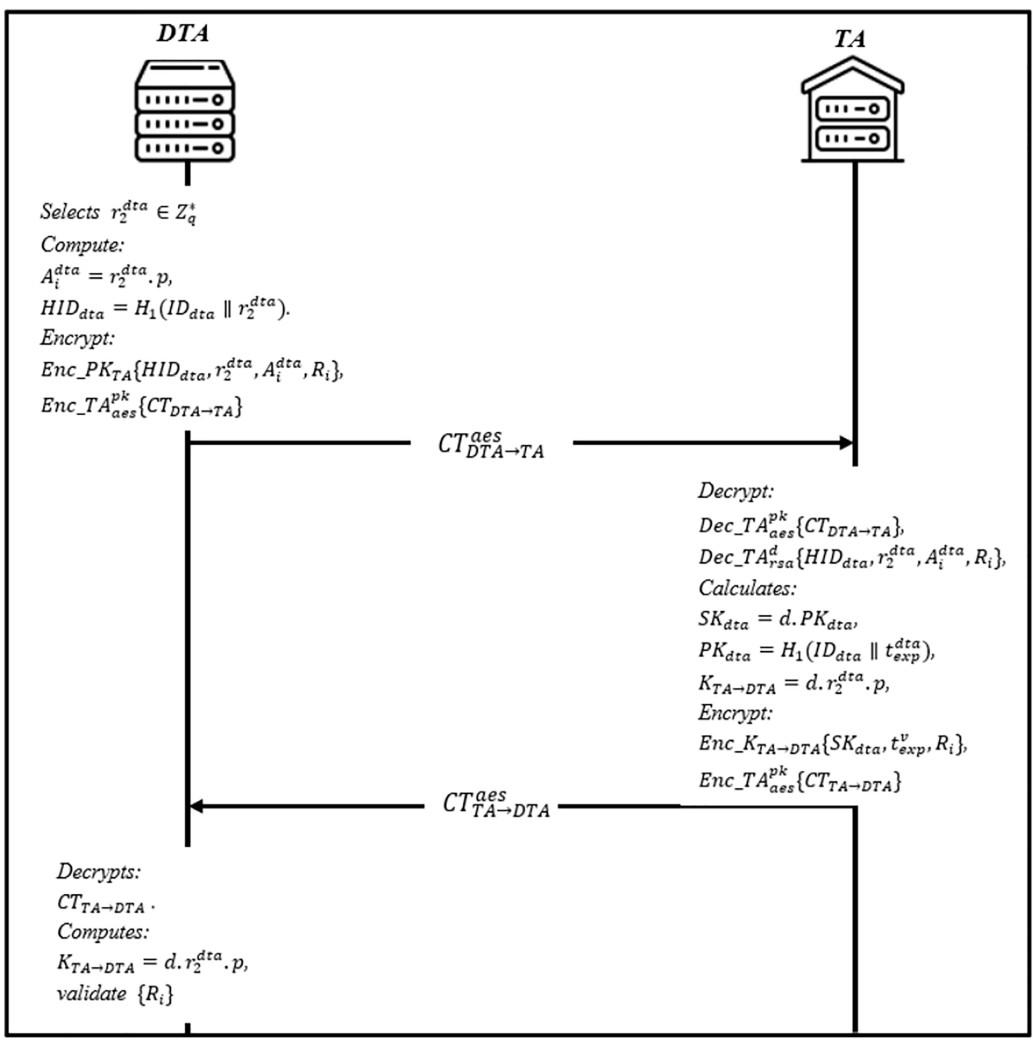

**Figure 5 Domain trusted authority registration phase.**

current phase. Then, TA checks whether the identity already exists in the database or not; if yes, send a notification; otherwise, apply the following steps:

1. Firstly, DTA selects a random number $r_2^{dta} \in Z_q^*$ as a secret key and compute $A_i^{dta} = r_2^{dta}.p, and HID_{dta} = H_1(ID_{dta} \| r_2^{dta})$. Then encrypt the hashed identity with RSA's public key $Enc\_PK_{TA}\{HID_{dta}, r_2^{dta}, A_i^{dta}, R_i\}$, to get the ciphertext $CT_{DTA \rightarrow TA} = Enc\_PK_{TA}HID_{dta}, r_2^{dta}, A_i^{dta}, R_i$, where $R_i$ is the secret value. The AES's public key is then utilized to encrypt the ciphertext $CT_{DTA \rightarrow TA}$ to get $CT_{DTA\beta TA}^{aes} = Enc\_TA_{aes}^{pk}\{CT_{DTA \rightarrow TA}\}$. DTA sends $CT_{DTA \rightarrow TA}^{aes}$ to TA.

2. When TA receives $CT_{DTA \rightarrow TA}^{aes}$, it will first decrypt $Dec\_TA_{aes}^{pk}\{CT_{DTA \rightarrow TA}\}$, and then decrypt the ciphertext $Dec\_TA_{rsa}^d HID_{dta}, r_2^{dta}, A_i^{dta}, R_i$ using it is the private key to obtain $<HID_{dta}, r_2^{dta}, A_i^{dta}, R_i>$, it also calculates it is a private key $SK_{dta} = d.PK_{dta}$, where

$PK_{dta} = H_1(ID_{dta}\|t_{exp}^{dta})$ is the public key of DTA, and $t_{exp}^{v}$ is the expiration of $SK_{dta}$. TA calculates the shared session key with DTA $K_{TA\beta DTA} = d.r_2^{dta}.p$ and encrypt the parameters $< SK_{dta}, t_{exp}^{v}, R_i >$ with the session key $CT_{TA\beta DTA} = Enc\_K_{TA\beta DTA} SK_{dta}, t_{exp}^{v}, R_i$. Finally, the ciphertext is further encrypted with AES public for secure communication $CT_{TA \to DTA}^{aes} = Enc\_TA_{aes}^{pk}\{CT_{TA \to DTA}\}$, and sends $CT_{TA \to DTA}^{aes}$ to DTA.

3. Upon receiving $CT_{TA \to DTA}^{aes}$ from TA, DTA decrypts it using AES public key and then decrypts $CT_{TA \to DTA}$. DTA computes $K_{TA\beta DTA} = d.r_2^{dta}.p$ to obtain $SK_{dta}, t_{exp}^{v}, R_i$. DTA then validate the $R_i$, if valid, DTA stores $SK_{dta}, t_{exp}^{v}$; otherwise, DTA rejects it.

## RSU registration phase

All RSUs submit their registration information to DTA within their domain area. Before the RSU registration phase, the DTA select a group private/public key that only valid in this area based on RSA key generation $sk_{dta}^{'} = r_2^{dta}$, and $pk_{dta}^{'} = r_2^{dta}.p$. Then DTA uses the private key $sk_{dta}^{'}$ to generate signature $Sign_{dta} = Sign\_sk_{dta}\{HID_{dta}, t_{exp}^{dta}, pk_{dta}^{'}\}$. DTA also calculates $X_{dta} = r_2^{dta}.pk_{dta}^{'}, I_{dta} = X_{dta} + H_2(M_{dta}^{'}, X_{dta})$ where $M_{dta}^{'}$ is $M_{dta}^{'} = HID_{dta}\|t_{exp}^{dta}\|pk_{dta}^{'}\|r_2^{dta}$. The DTA then concatenated the signature with the message $CT_{DTA \to RSU} = Enc\_DTA_{DTA \to RSU}^{aes}\{Sign_{dta}\|M_{dta}^{'}\}$, and broadcasting $CT_{DTA \to RSU}$ to the RSUs in this domain. Upon receiving $CT_{DTA \to RSU}$, RSU decrypts it $Dec\_DTA_{DTA\beta RSU}^{aes}$ $\{Sign_{dta}\|M_{dta}^{'}\}$ to obtain the parameters and compute the public key based on domain identity and expiration time $pk_{dta} = H_1(HID_{dta}\|t_{exp}^{dta})$. The RSU validates the $Sign_{dta}$ by comparing it with new computed signature $Sign_{dta}^{'} \neq Sign_{dta}$, if valid, stores $HID_{dta}, t_{exp}^{dta}, pk_{dta}^{'}$ and apply the registration steps and as shown in Fig. 6.

1. The RSUs generates a random number $r_{rsu} \in Z_q^*$ as a secret key and computes $A_i^r su = r_r su.p$, and $RID_{rsu} = H_1(ID_{rsu}\|r_{rsu})$. RSU encrypt the parameter RSA's public key $CT_{RSU \to DTA} = Enc\_PK_{DTA}\{RID_{rsu}, r_{rsu}\}, A_i^{rsu}, R_i$, where $R_i$ is the secret value. Then, generate ciphertext using AES's public key $CT_{RSU \to DTA}^{aes} = Enc\_DTA_{aes}^{pk}\{CT_{RSU \to DTA}\}$, and sends $CT_{RSU \to DTA}^{aes}$ to DTA.

2. Upon receiving $CT_{RSU \to DTA}^{aes}$, DTA decrypts is using $Dec\_DTA_{aes}^{pk}\{CT_{RSU \to DTA}\}$, and also decrypts $Dec\_DTA_{rsa}^{d}\{RID_{rsu}, r_{rsu}, A_i^{rsu}, R_i\}$ to get $< RID_{rsu}, r_{rsu}, A_i^{rsu}, R_i >$. DTA generates a RSU's private key $SK_{rsu} = r_2^{dta}.PK_{rsu}$, where $PK_{rsu} = H_1(RID_{rsu}.r_{rsu})$. Then, it calculates the session key with DTA $K_{DTA \to RSU} = r_2^{dta}.r_{rsu}.p$, and $CT_{DTA \to RSU}$ : $Enc\_K_{DTA \to RSU}\{SK_{rsu}, t_{exp}^{rsu}, R_i + 1\}$, where $t_{exp}^{rsu}$ is the expiration of $SK_{rsu}$. The ciphertext is further encrypted with AES's public key $CT_{DTA \to RSU}^{aes} = Enc\_RSU_{aes}^{pk}\{CT_{DTA \to RSU}\}$, and sends $CT_{DTA \to RSU}^{aes}$ to RSU.

3. After receiving the RSU decrypts $Dec\_RSU_{aes}^{pk}\{CT_{DTA \to RSU}\}$, to obtain $CT_{DTA \to RSU}$ and compute session key with DTA $K_{DTA \to RSU} = r_2^{dta}.PK_{dta}$ and decrypts $Dec\_K_{DTA \to RSU}\{SK_{rsu}, t_{exp}^{rsu}, R_i + 1\}$, to get $< SK_{rsu}, t_{exp}^{rsu}, R_i + 1 >$ if valid, stores $SK_{rsu}, t_{exp}^{rsu}$. Otherwise, RSU rejects it.

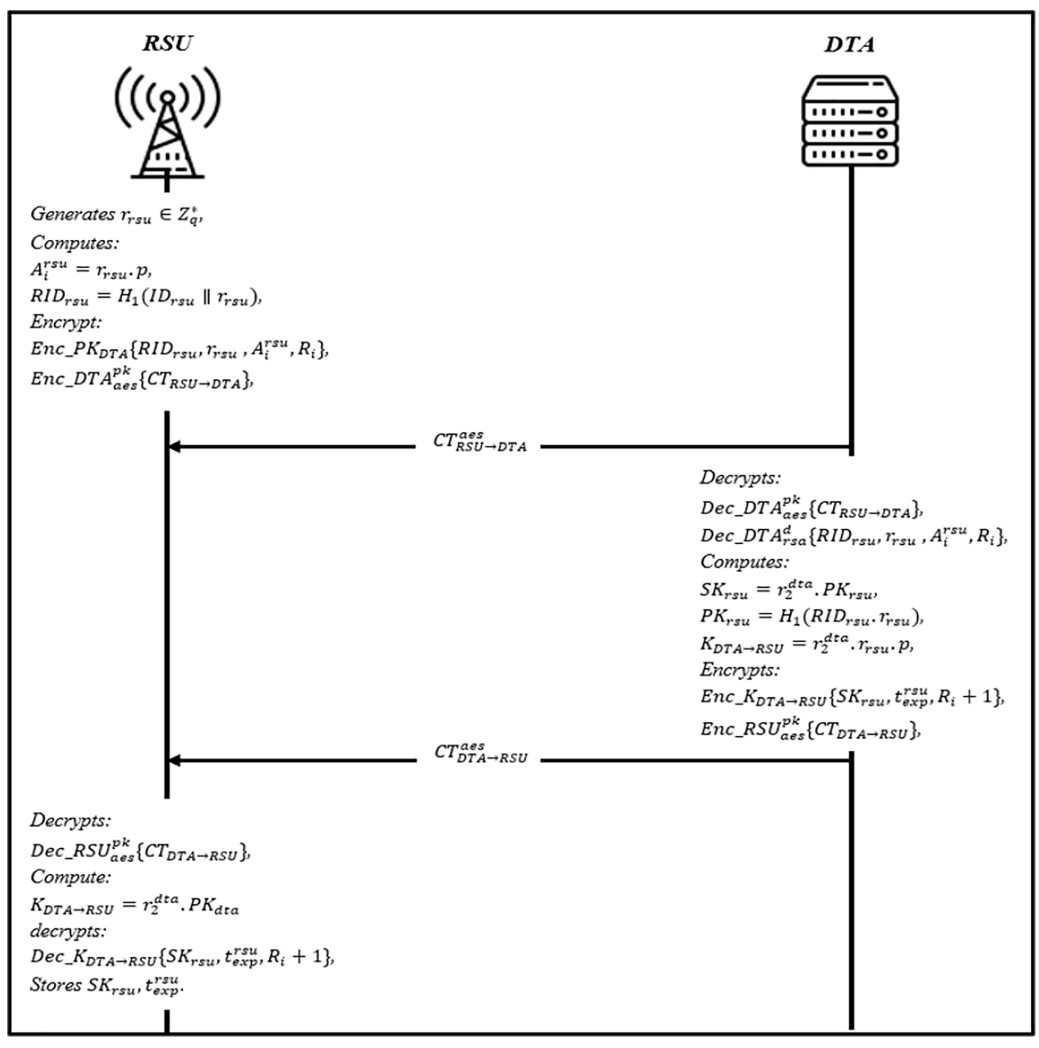

**Figure 6** RSU registration phase.

## Online joining phase

In this phase, the vehicle will send a joining request to the DTA through the RSU. The information is broadcasted to each vehicle within the domain to enable the vehicle to get authenticated. The joining steps are shown in Fig. 7 and described as follow:

1. The RSU1 broadcasts $ID_{rsu1}, t_{exp}^{dta}, t_{exp}^{rsu}, T_1, R_i, ID_{dta}, PK_{dta}, Sign_{rsu1})$ and $Sign_{dta}$ regularly, where $Sign_{rsu1}) = Sign\_sk_{rsu1})\{ID_{rsu1}), ID_{dta}, t_{exp}^{rsu}, T_1, R_i\}$, and calculates $X_{rsu1} = r_2^{rsu}.pk_{rsu1})', I_{rsu1} = X_{rsu1} + H_2(M_{rsu1})', X_{rsu1})$, and $M_r su' = ID_{rsu1}\|ID_{dta}\|t_{exp}^{rsu1}\| T_1\|R_i$. Then, it encrypts it using AES public key $CT_{RSU\to V} = Enc\_V_{RSU\to V}^{aes} \{Sign_{rsu1}\|M_{rsu}'\}$, and sends $CT_{RSU} \to_V$ to the vehicle.

2. Upon receiving, Vehicle decrypt $CT_{RSU} \to_V$ using the public key $Dec\_V_{RSU\to V}^{aes} \{Sign_{rsu1})\|M_{rsu}'\}$ to get the signature. Then, it computes $pk_{dta} = H_1(HID_{dta}\|t_{exp}^{dta}$ and verifies $Sign_{rsu1})$, if invalid, end the session; otherwise, the vehicle continues to verify the freshness of the timestamp $T_1$ and validity of the $Sign_{rsu1})$, if validation successful,

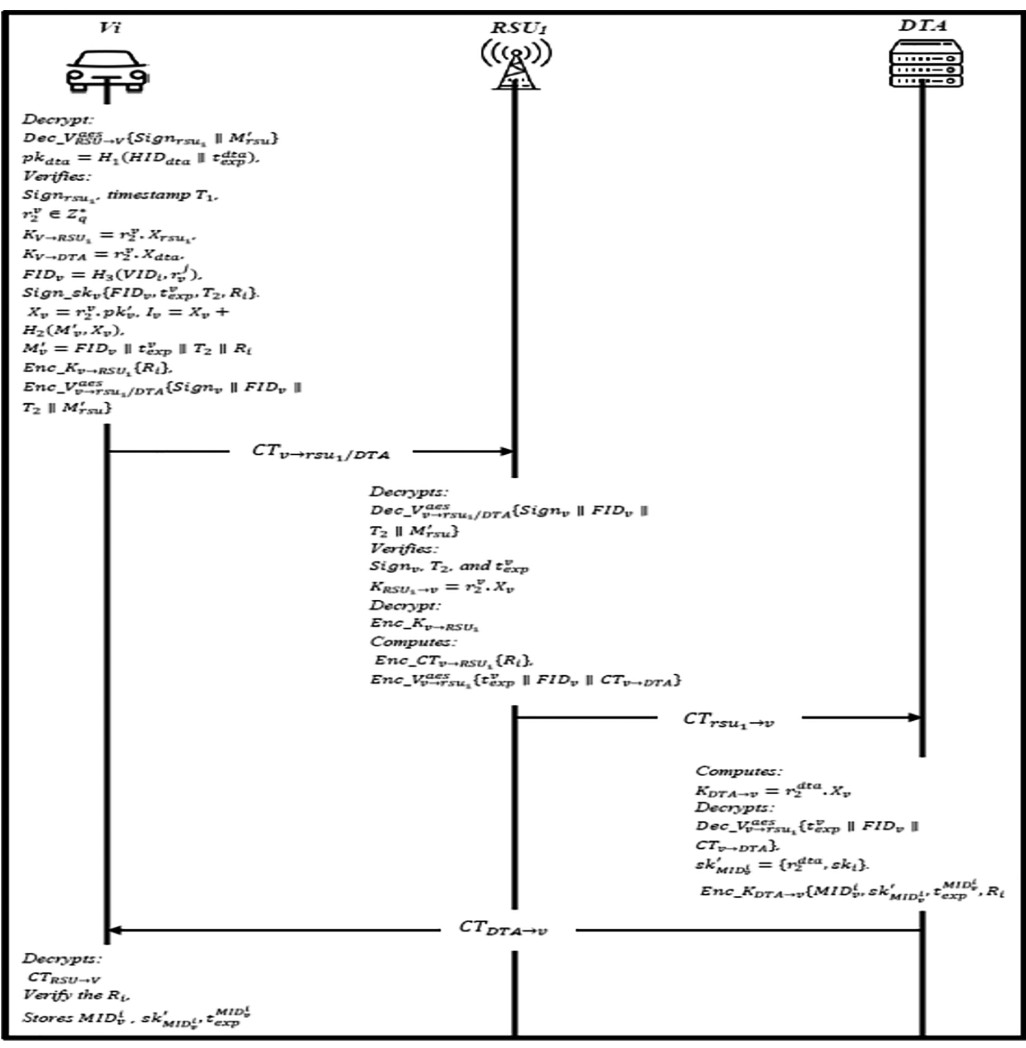

**Figure 7 Online joining phase.**

DTA and RSU1 are considered legal entities. Vehicle choose a random number $r_2^v \in Z_q^*$ and compute session key with RSU1 $K_{(V \to RSU_1)} = r_2^v.X_{rsu1})$ and the session key with DTA $K_{V \to DTA} = r_2^v.X_{dta}$ respectively. The vehicle finally choose pseudonyms $FID_v = H_3(VID_i, r_v^j)$ and generates the signature $Sign_v = Sign_{skv}\{FID_v, t_{exp}^v, T_2, R_i\}$. It also calculates $X_v = r_{2^v}.pk_{v'}, I_v = X_v + H_2(M(v)', X_v)$, and $M_v' = FID_v \| t_{exp}^v \| T_2 \| R_i$ and encrypts the secret value $Enc\_Kv \to RSU1 \| R_i$, and $Enc\_K_v \to DTA \| R_i$. Then AES public utilized to encrypt the message $CT_{v \to rsu_1/DTA} = Enc\_V_{v \to rsu_1/DTA}^{aes} \{Sign_v \| FID_v \| T_2 \| M_{rsu}'\}$ to RSU1.

3. When the RSU1 receives the message, it decrypts the $Dec\_V_{v \to rsu_1/DTA}^{aes}\{Sign_v \| FID_v \| T_2 \| M_{rsu}'\}$ and verifies $Sign_v$, $T_2$, and $t_{exp}^v$ accordingly. If the verification goes well, RSU1 generates a shared session key $K_{RSU_1 \to v} = r_2^v.X_v$ to decrypt $Enc\_Kv \to RSU_1$ and check the validity of $R_i$. Finally computes $CT_v \to DTA = Enc\_CTv \to RSU_1\{R_i\}$ and sends $CT_{rsu_1 \to v} = Enc\_V_{v \to rsu_1}^{aes}\{t_{exp}^v \| FID_v \| CT_{v \to DTA}\}$ to DTA.

4. Upon receiving the message, DTA computes the session key $K_{DTA \to v} = r_2^{dta}.X_v$ and decrypts $Dec\_V_{v \to rsu_1}^{aes}\{t_{exp}^v \| FID_v \| CT_{v \to DTA}\}$ and also decrypt $CT_{(v \to DTA)}$ to get $R_i$. If valid, DTA generates a group of identities $MID_v^i$ and the group private key $sk_{MID_v^i}' = r_2^{dta}, sk_i$ for the vehicle. The DTA encrypt the message using the session key $CT_{DTA \to v} = Enc\_K_{DTA \to v}\{MID_v^i, sk_{MID_v^i}', t_{exp}^{MID_v^i}, R_i\}$, where $t_{exp}^{MID_v^i}$ is expiration of $MID_v^i$. The DTA sends $CT_{DTA \to v}$ to RSU1, and RSU1 forwards the $CT_{DTA \to v}$, and $CT_{RSU \to V}$ to vehicle.

5. The vehicle decrypts the $CT_{RSU \to V}$ and verify the secret value $R_i$, if valid, then a secure channel is established. The $MID_v^i, sk_{MID_v^i}', t_{exp}^{MID_v^i})$, and $R_i$ is obtained now after decryption, and vehicle stores $MID_v^i, sk_{MID_v^i}', t_{exp}^{MID_v^i}$.

## Online crossover phase

When the vehicle crosses from one domain to another, it needs to send a joining request to the RSU2 located in different geographical domains. After the RSU2 broadcasted the information to each vehicle, it will send an authentication message to RSU2, where this phase is called the crossover phase. Figure 8 shows the steps of this phase and described as follows:

1. The RSU2 broadcasts $ID_{rsu_2}, t_{exp}^{rsu_2}, T_3, R_i, Sign_{rsu_2}$ and $Sign_{dta}$ regularly, where $Sign_{rsu_2} = Sign\_sk_{rsu_2}\{ID_{rsu_2}, t_{exp}^{rsu_2}, T_3, R_i\}$, and calculates $X_{rsu_2} = r_2^{rsu_2}.pk_{rsu_2}', I_{rsu_2} = X_{rsu_2} + H_2(M_{rsu_2}', X_{rsu_2}$, and $M_{rsu_2}' = ID_{rsu_2} \| t_{exp}^{rsu_2} \| T_3 \| R_i$. Then, it encrypts it using AES public key $CT_{RSU \to V} = Enc\_V_{RSU \to V}^{aes}\{Sign_{rsu_2} \| M_{rsu_2}'\}$, and sends $CT_{RSU} \to v$ to the vehicle.

2. The vehicle gets the message and decrypts it using AES's public key $Dec\_V_{RSU \to V}^{aes}\{Sign_{rsu_2} \| M_{rsu_2}'\}$ to obtain a signature, then it verifies the $T_3$ whether is fresh or not, if not, end the session. Otherwise, the vehicle generates a shared session key with RSU2 $K_{V \to RSU_2} = r_2^v.X_{rsu_2}, G\_Sign\_SK_{MID_v^i}\{MID_v^i, T_4, t_{exp}^{MID_v^i}, R_i\}, X_{rsu_2} = r_2^{rsu_2}.pk_{rsu_2}', I_{rsu_2} = X_{rsu_2} + H_2(M_{rsu_2}', X_{rsu_2})$, and $M_{rsu_2}' = ID_{rsu_2} \| ID_{dta} \| t_{exp}^{rsu_2} \| T_4 \| R_i$. Then, it encrypts it using AES public key $CT_{V \to RSU_2} = Enc\_V_{V \to RSU_2}^{aes}\{Sign_v \| M_{rsu_2}'\}$, and sends $CT_{V \to RSU_2}$ to the RSU2.

3. The RSU2 first decrypts $Dec\_V_{V \to rsu_2}^{aes}\{Sign_v \| M_{rsu_2}'\}$, and verifies the timestamp $T_4$, and signature $Sign_v$ by computing the public of the vehicle $pk_{MID_v^i} = H_1(MID_v^i \| t_{exp}^{MID_v^i})$, if invalid, end session; otherwise, vehicle $MID_v^i$ is legal. Finally, RSU2 generates a shared session key with the vehicle $K_{RSU_2 \to v} = r_2^{rsu_2}.X_v$, and compute $CTRSU_2 \to v = Enc\_kRSU_2 \to v\{R_i\}$, then encrypt the ciphertext using AES public key $Enc\_V_{rsu_2 \to v}^{aes}\{CT_{rsu_2 \to v})$, and send it to the vehicle.

4. The vehicle uses the AES public key to decrypt the message $Dec\_V_{rsu_2 \to v}^{aes}\{CT_{rsu_2 \to v}\}$, to obtain $CTrsu_2 \to v$ to decrypt it using a shared session key $K_{v \to rsu_2} = r_2^{rsu_2}.X_{rsu_2}$, if the secret value $R_i$ is valid, then a trust relationship is created; otherwise, authentication fails.

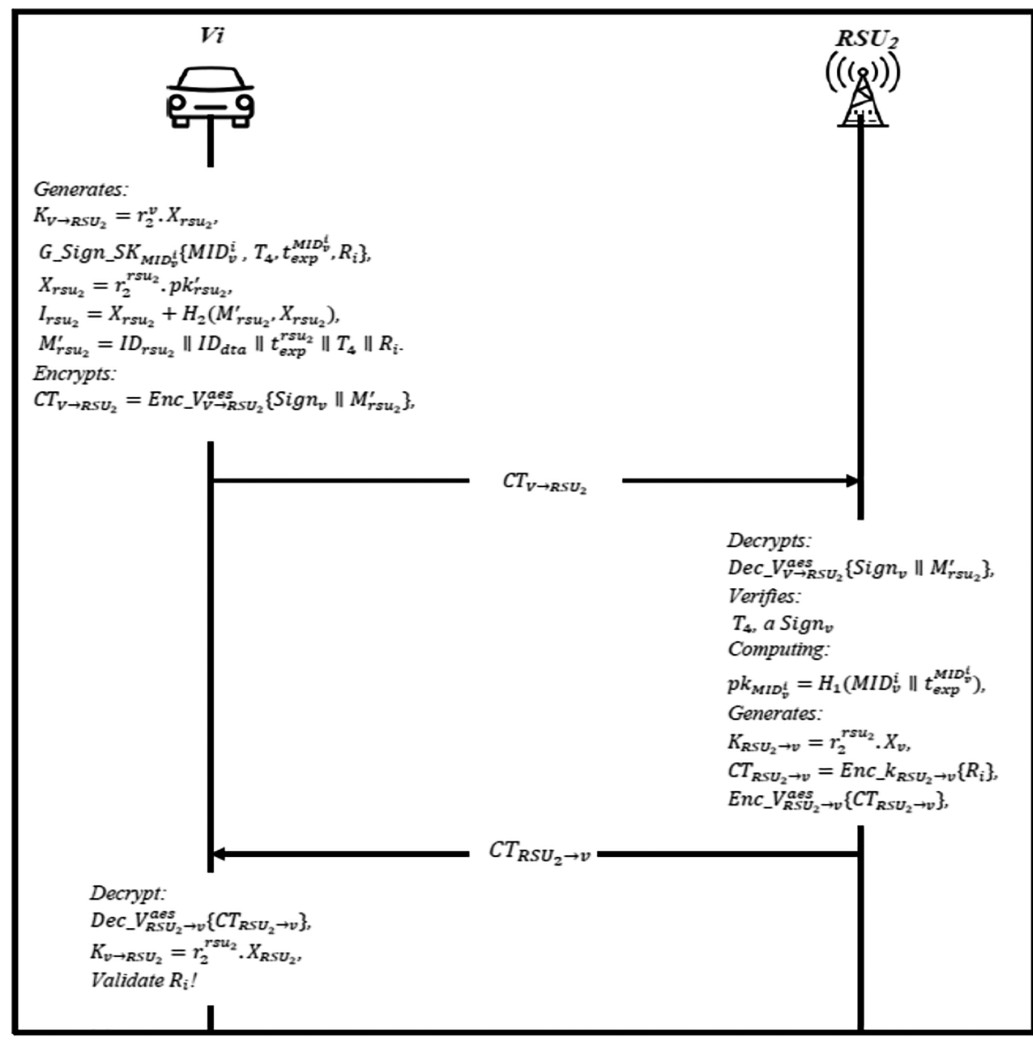

**Figure 8 Online crossover phase.**               

## Offline crossover phase

As the secret credentials have been preloaded priorly into the RSUs, the movement from RSU1 to RSU2 does occur dynamically. Therefore, when the vehicle leaves RSU1, crossover authentication is required to execute. The following steps are described as follows:

1. The RSU2 preloads the parameters $r_v^j, SK_{rsu_2}^j, t_{exp}, R_i, TID_v, ID_{rsu_2}, t_{exp}^{dta}, t_{exp}^{rsu}, T_1, Sign_{rsu_2}$, where the $Sign_{rsu_2} : Sign\_SK_{rsu_2}\{ID_{rsu_2}, t_{exp}^{rsu_2}, T_2, R_i, TID_v, r_{rsu_2}\}$, where $t_{exp}^{rsu_2}$ is the expiration of $SK_{rsu_2}$, and $r_{rsu_2} \in Z_q^*$ is a random number. The RSU2 encrypts the offline signature using AES public key $CTrsu_2 \rightarrow v : \{Sign_{rsu_2}\}$ and sends $CT_{rsu_2} \rightarrow v$ to vehicle.

2. Upon receiving $CT_{rsu_2} \rightarrow v$, vehicle decrypts it using the public key to get the offline signature $Sign_{rsu_2}$, then decrypt the signature using the private key of the vehicle to obtain $< ID_{rsu_2}, t_{exp}^{rsu_2}, T_2, R_i, TID_v, r_{rsu_2} >$. The vehicle verifies the timestamps $T_2$, if not fresh, authentication failed; otherwise, the vehicle generates a shared session key

$K_{(v \rightarrow rsu_2)} = r_2^v . X_{rsu_2}$ and select a unique private key to sign $ID_{rsu_2}, t_{exp}^{rsu_2}, T_2, R_i, TID_v,$ $r_{rsu_2} >$ , $Sign_{rsu_2} : Sign\_SK_{rsu_2}\{ID_{rsu_2}, t_{exp}^{rsu_2}, T_2, R_i, TID_v, r_{rsu_2} >, \}$, and then it encrypts the signature using AES public key $CTv \rightarrow rsu_2: \{Sign_{rsu_2}\}$ and sends $CTv \rightarrow rsu_2$ to RSU.

3. After receiving $CTv \rightarrow rsu_2$ from the vehicle, RSU2 decrypts it using AES public key to obtain the signature $Sign_{rsu_2}$, then use the RSU2 private key to get the parameters $t_{exp}^{rsu_2}, T_2, R_i, TID_v, r_{rsu_2}$. RSU2 verifies the $t_{exp}^{rsu_2}, R_i, and \ T_2$, if verification is not equal, end session. Otherwise, generate a shared session key with the vehicle $K_{rsu_2 \rightarrow v} = r_2^v . X_{rsu_2}$, and compute $CT\_K_{rsu2} \rightarrow v \{R_i\}$ and sends $CT\_K_{rsu2 \rightarrow} v$ to vehicle.

4. The vehicle receives the message using $CT\_K_{rsu2} \rightarrow v \{R_i\}$, if the secret value is not matched, terminate the session. Otherwise, an offline authentication is established between the vehicle and RSU2.

## SECURITY ANALYSIS

We provide a complete overview of the proposed scheme's security in this section to illustrate how the proposed scheme has provided robust security. The study was carried out using Burrows, Abadi, and Needham's logic in our scheme to demonstrate mutual authentication between the vehicle and other participating entities (BAN). Finally, in this section, a theoretical security examination, called informal analysis, has been discussed.

### Informal analysis

The proposed scheme's security has been discussed in this sub-section in an informal review to show that the scheme provides strong security protection for associated security properties and attacks. We justify the defence of the device and attacks in the following terms of security properties. Table 3 shows the comparison of the security features of the proposed scheme against other schemes.

1. **Message Integrity and authentication:** In the proposed scheme, a hash function $h(\cdot): 0,1 \ ^* \rightarrow Z \ ^* \ q$ is utilized to the message signature that makes the faking of the message is impossible. To generate the signature, the message is further attached with secret key of the RSA algorithm to the hashed value of the message, e.g., $Sign_{dta} = Sign\_sk_{dta}\{HID_{dta}, t_{exp}^{dta}, pk_{dta}'\}$ by the sender. Upon receiving, the receiver can decode the message and check its validity by comparing it with the latest computed message and the RSUs. DTA can effectively ensure the message's integrity. Therefore, message integrity and authentication are supported by the proposed scheme.

2. **Message unforgeability:** The proposed scheme is achieved by $Sign_{dta}$, and $h(\cdot)$. The trusted authority generates the signature with a private key d, and this key is held secretly by the TA. The attacker is, therefore, cannot compute the session key that shared between entities and TA; the session $K_{TA \rightarrow v} = d.A_i$ is based on the secret key of the TA, and the attacker cannot forge the message. Also, the exchanged messages are further encrypted using the AES public key for secure communication; thus, the attacker cannot obtain the secret value $R_i$ of the entity. Therefore, only the specified

**Table 3 Comparison of security features.**

| | ID-CPPA *Ali & Li (2020)* | AAAS *Jiang, Ge & Shen (2020)* | HCDA (*Tan, Xuan & Chung (2020)*) | Proposed scheme |
|---|---|---|---|---|
| Message integrity and authentication | ✓ | ✓ | × | ✓ |
| Message unforgeability | × | × | ✓ | ✓ |
| Identity privacy-preserving | ✓ | ✓ | ✓ | ✓ |
| Non-repudiation | × | × | × | ✓ |
| Unlinkability | ✓ | ✓ | × | |
| Forward secrecy | × | ✓ | × | ✓ |
| Cross-domain Property | ✓ | ✓ | ✓ | ✓ |
| Offline authentication | × | × | × | ✓ |
| Impersonation Attacks | ✓ | × | ✓ | ✓ |
| Modification attack | ✓ | ✓ | ✓ | ✓ |
| Reply attack | ✓ | ✓ | ✓ | ✓ |
| Man-in the middle attack | ✓ | × | × | ✓ |
| Brute-force attack | × | × | × | ✓ |

RSUs, can obtain $R_i$, and the proposed scheme can protect the message from being forged and generate the related hash function.

3. **Identity privacy-preserving:** The pseudonyms $FID_v = H_3(VID_i, r_v^j), HID_{dta} = H_1(ID_{dta}\|r_2^{dta})$, and $RID_{rsu} = H_1(ID_{rsu} \| r_{rsu})$ are hashed along with identity and the random number; hence, the adversaries cannot obtain the vehicle's real identity and RSUs. Furthermore, it used to calculate several parameters $T_i = H(VID_i\|s), PK_{dta} = H_1(ID_{dta}\|t_{exp}^{dta})$, and $M_{dta}' = HID_{dta}\|t_{exp}^{dta}\|pk_{dta}'\|r_2^{dta}$ the attacker cannot obtain the real identity because the identity is secured using a one-way hash function. Also, in each communication session, the pseudonyms used are different, so no opponent can obtain the identity and trace the vehicle from the message it sends. Therefore, identity and location privacy is achieved by the proposed scheme.

4. **Non-repudiation:** In the proposed scheme, the messages $CT_{RSU \to V}, Enc\_K_{v \to DTA}\{R_i\}$, and $CT_{DTA \to v}$ contains different values, *e.g.*, $\{Sign_v\|FID_v\|T_2\|M_{rsu}'\}$, where $M_{rsu}' = ID_{rsu_1}\|ID_{dta}\|t_{exp}^{rsu_1}\|T_1\|R_i$, it has the secret value $R_i$ that know to RSUs, and DTA, the vehicle cannot deny the message it has received because of the secret value. The freshness of the timestamps also plays a vital role in checking the validity of the message. Therefore, the proposed scheme achieved the non-repudiation property.

5. **Unlinkability:** The message $ID_{rsu_1}, t_{exp}^{dta}, t_{exp}^{rsu}, T_1, R_i, ID_{dta}, PK_{dta}, Sign_{rsu_1}$ in each broadcasting operation, the RSUs are transmitted, which is different. Also, the secret $SK_rsu$ is valid only for one session communication. Furthermore, the identity of the vehicle is further secured with a one-way hash function. Therefore, the adversary cannot expect that messages belong to the same vehicle. Thus, the proposed scheme provides desired unlinkability.

6. **Forward secrecy:** In the proposed scheme, the broadcasted parameters $ID_{rsu_1}, t_{exp}^{dta}, t_{exp}^{rsu}, T_1, R_i, ID_{dta}, PK_{dta}, Sign_{rsu1}$ indicates the legitimacy of the entity's identities. All these broadcasted parameters do not contain information about other credentials of the vehicles. Also, the session keys are used only for a single session to communicate, and although that the message is encrypted with these short-lived keys, the keys are further encrypted with AES public key. Consequently, attackers cannot obtain any information about other credentials. Therefore, the proposed scheme provides perfect forward secrecy.

7. **Cross-domain Property:** According to the proposed scheme's specification, two vehicles belong to different domains and are separately registered with domain trusted authorities. Every domain trusted authority has separate RSUs with vehicles and can connect mutually through the domain trusted authority.

8. **Offline Authentication:** In the proposed scheme, TA preloads the credentials $r_v^j, SK_v^j, t_{exp}, R_i, TID_v$ in RSUs priorly in their domain. Then, RSU1 preloads $ID_{rsu_1}, t_{exp}^{dta}, t_{exp}^{rsu}, T_1, R_i, ID_{dta}, PK_{dta}, Sign_{rsu_1}$ into the vehicles in prior deployment. This helps the vehicle to authenticate to the domain in offline mode while the connectivity is temporarily unavailable. Therefore, the proposed scheme provides an offline authentication.

9. 9. **Impersonation Attacks:** If the adversary impersonate one of the registered vehicles or RSUs, it should construct a message $ID_{rsu_1}, t_{exp}^{dta}, t_{exp}^{rsu}, T_1, R_i, ID_{dta}, PK_{dta}, Sign_{rsu_1}$ to meet the verification process. However, it will be difficult for the adversary to pass the verification because the signature is generated using the public key of the entity, and the parameters $M_{rsu}' = ID_{rsu_1} \| ID_{dta} \| t_{exp}^{rsu_1} \| T_1 \| R_i$ are concatenated with signature and encrypted using the public key $CT_{RSU\|V} = Enc\_V_{RSU \rightarrow V}^{aes} \{ Sign_{rsu_1} \| M_{rsu}' \}$. The message also contains a secret Ri value that the recipient verifies to verify the message's validity. Therefore, no impersonation attack on the current technique can be launched by the adversary.

10. **Modification attack:** Assume the adversary get the encryption key during the transmission and modify the message $Enc\_V_{RSU \rightarrow V}^{aes} \{ Sign_{rsu_1} \| M_{rsu}' \}$, he/she will not be able to obtain the signature values $ID_{rsu_1}, ID_{dta}, t_{exp}^{rsu}, T_1, R_i$ because it is encrypted using the secret key of the vehicle or RSUs. Also, the adversary will not pass the verification process because the message cannot be decrypted. However, the receiver who has the private key and the secret value stored in the initial registration phase is used to check the message's validity. Therefore, the proposed scheme withstands the modification attack.

11. **Reply attack:** In the proposed scheme, a timestamp is used in every message, *e.g.*, $M_{rsu}' = ID_{rsu_1} \| ID_{dta} \| t_{exp}^{rsu_1} \| T_1 \| R_i$ has the timestamp of the current session, and respectively after receiving, the freshness of the timestamp will be validated by comparing it with the current timestamp $T_1 \neq \Delta T$ of the system. Furthermore, the shared session key between entities has an expiration time, *e.g.*, $t_{exp}^{rsu_1}$, and $t_{exp}^{dta}$. Therefore, the proposed scheme resistance to reply attacks.

12. **Man-in-the-middle attack:** The transmitted messages may be intercepted, and the adversary could do a particular modification. In the proposed scheme, the secret vehicle key $s \in Z_q^*$, is generated randomly; also, the $T_i = H(VID_i \| s)$, is computed based on the random number. The secret value Ri is generated randomly, sent alongside the message, and encrypted using the vehicle private key to create the signature. So, the message is transmitted in encrypted form, and it will be difficult for the adversary to get this information. Besides, if the attacker intercepts the message, the receiving message will be delayed, and it will not pass the validation process due to the timestamp usage $T_1 T$. The proposed scheme, therefore, withstands the man-in-the-middle attack.

13. **Brute-force attack:** In our scheme, various generated random, *e.g.*, $s \in Z_q^*, r_2^{dta} \in Z_q^*$, and $r_{rsu} \in Z_q^*$ are used to secure the identities and sent securely to the vehicle or RSUs by encrypting them using AES public key and RSA key. If the adversary wants to break the authentication message, he/she needs to know the secret vehicle parameters or identity $VID_i$. But, in the proposed scheme, the identity is secured using a one-way hash function and concatenated with random number $T_i = H(VID_i \| s)$. Then, this hash value is encrypted using RSA $Enc\_TA_{rsa}^{pk}\{T_i\}$, to find the value, the adversary will try all the numbers (brute-force) till find the value which transmission will be delayed and results in authentication fails due to the timestamp. So, the chance of the adversary to get/brute-force the correct value is infinitesimal. Therefore, the proposed scheme has resistance to a brute-force attack.

## Burrows, abadi, and needham (BAN) logic

We use Burrows, Abadi, and Needham BAN logic in this subsection, which is used to prove the correctness of authentication methods, beginning with the solution's formalization, followed by postulates to achieve the objectives emphasized. Nonetheless, with the commonly used BAN logic technique, we show the mutual authentication validity between the vehicle and RSU. In the BAN logic analysis, Table 4 displays the related notations. We start by explaining the notes used to do the demonstration, followed by BAN logic postulates, followed by the formal idealization of the scheme's messages; we also list the assumptions of the solution and highlight the goals.

    **Security Goals:** This process shows the session key authentication goals between vehicles and RSU that authenticated mutually. Thus, there five goals primarily used in the proposed scheme and established as follows:

- **Goal 1:** $DTA| \equiv V_i| \equiv (VID_i)$.
- **Goal 2:** $DTA| \equiv (VID_i)$.
- **Goal 3:** $DTA| \equiv RSU| \equiv (RID_{rsu})$.
- **Goal 4:** $DTA| \equiv (RID_{rsu})$.
- **Goal 5:** $RSU| \equiv DTA| \equiv (k_{dta \to rsu})$.
- **Goal 6:** $RSU| \equiv (k_{dta \to rsu})$.
- **Goal 7:** $V_i| \equiv RSU| \equiv (pk_{dta}')$.

**Table 4 Notation and description in BAN logic.**

| Notation | Description |
|---|---|
| $P\vert\equiv B$ | $P$ believes $B$ |
| $\#(B)$ | $B$ is fresh |
| $P\Rightarrow B$ | $P$ has jurisdiction over $B$ |
| $P\triangleleft B$ | $P$ sees $B$ |
| $P\vert\sim B$ | $P$ once said $B$ |
| $(B,Y)$ | $B$ or $Y$ is one part of $(B,Y)$ |
| $\langle B\rangle_Y$ | $B$ combined with $Y$ |
| $(B)_Y$ | $B$ is fresh with the key $K$ |
| $P\overset{K}{\leftrightarrow}Q$ | $P$ and $Q$ use the shared key $K$ to communicate |
| **SK** | The current session key |
| $\dfrac{P\vert\equiv P\overset{k}{\leftrightarrow}Q,\ P\triangleleft\{B\}_k}{P\vert\equiv Q\vert\sim B}$ | The message-meaning rule |
| $\dfrac{P\vert\equiv\#(B)}{P\vert\equiv\#(B,Y)}$ | The freshness-conjuncatenation rule |
| $\dfrac{P\vert\equiv\#(B),\ P\vert\equiv Q\vert\sim B}{P\vert\equiv Q\vert\equiv B}$ | The nonce verification |
| $\dfrac{P\vert\equiv Q\Rightarrow B,\ P\vert\equiv Q\vert\equiv B}{P\vert\equiv B}$ | The jurisdiction rule |

- **Goal 8:** $V_i\vert\equiv(pk'_{dta})$.

**Messages:** In this process, we idealize the scheme phase to represent the exchanged messages between the main entities of the scheme; the message representation is shown as follows:

- **Msg$_1$:** $V_i\to RSU:\{Sign_v\|FID_v\|T_2\|M'_{rsu}\}$.
- **Msg$_2$:** $RSU\to DTA:\{t^v_{exp}\|FID_v\|CT_{v\to DTA}\}$.
- **Msg$_3$:** $DTA\to RSU:\{t^v_{exp}\|FID_v\|CT_{v\to DTA}\}$.
- **Msg$_4$:** $RSU\to V_i:\{MID^i_v,sk'_{MID^i_v},t^{MID^i_v}_{exp},R_i\}$.

The messages of scheme can be idealized as follows:

- **SMI 1.** $V_i\rightarrowtail TA:(Sign_v)PK_{TA}$.
- **SMI 2.** $DTA\rightarrowtail TA:(ID_{dta})PK_{TA}$.
- **SMI 3.** $RSU\to DTA:(ID_{rsu})_{pk'_{dta}}$.
- **SMI 4.** $DTA\rightarrowtail RSU:(K_{DTA\beta RSU})(PK_{rsu})$.
- **SMI 5.** $RSU\to V_i:(pk_{MID^i_v})_{(h(MID^i_v)}$.

**Assumption:** The initialization situation of the proposed scheme depends on some assumptions to prove the scheme; the assumptions are as follow:

- **AS 1.** $TA\vert\equiv\#(T_1,R_i)$.

- **AS 2.** $DTA| \equiv \#(T_1, T_2, R_i)$.
- **AS 3.** $RSU| \equiv \#(T_3)$.
- **AS 4.** $V_i| \equiv \#(T_2, R_i)$.
- **AS 5.** $TA| \equiv |\xrightarrow{k_{TA \to v}} V_i$.
- **AS 6.** $DTA| \equiv |\xrightarrow{K_{DTA \to v}} V_i$.
- **AS 7.** $DTA| \equiv |\xrightarrow{K_{DTA \to RSU}} RSU$.
- **AS 8.** $V_i| \equiv V_i \xleftarrow{VID} RSU$.
- **AS 9.** $DTA| \equiv V_i \Rightarrow (VID_i)$.
- **AS 10.** $DTA| \equiv RSU \Rightarrow (RID_{rsu})$.
- **AS 11.** $V_i | \equiv RSU \Rightarrow (SK_{rsu})$.
- **AS 12.** $RSU| \equiv |\xrightarrow{K_{DTA \to RSU}} DTA$.
- **AS 13.** $RSU| \equiv DTA \Rightarrow (K_{DTA \to RSU})$.

**Proof:** The stated security goals (Goal 1, Goal 2, Goal 3, Goal 4, Goal 5, Goals 6, Goal 7, and Goal 8) will be demonstrated in this process and achieved in this respect.

According to **SMI 1.** $V_{i \to} TA$: $(Sign_v)PK_{TA}$, we get:

**S1:** $TA \triangleleft (VID_i)K_{TA \to v}$.

From **S1, AS 4.** $V_i| \equiv \#(T_2, R_i)$, by utilizing message meaning ruling, we obtain:

**S2:** $DTA| \equiv V_i| \sim (VID_i)$.

From **S2, AS 1.** $TA| \equiv \#(T_1, R_i)$, and by utilizing the rule of freshness and nonce verification, we get:

**S3:** $DTA| \equiv V_i| \equiv (VID_i)$.

Thus, the **Goal 1:** $DTA| \equiv V_i| \equiv (VID_i)$ is achieved.

According to **S3:** $DTA|V_i|(VID_i)$, **AS 9.** $DTA| \equiv V_i \Rightarrow (VID_i)$., and by utilizing the rule of jurisdiction, we obtain:

**S4:** $DTA| \equiv (VID_i)$,

Thus, the **Goal 2:** $DTA| \equiv (VID_i)$, is achieved.

According to **SMI 2.** $DTA \to TA$: $(ID_{dta})PK_{TA}$, we have:

**S5:** $DTA \triangleleft (ID_{rsu})_{(pk'_{dta})}$

Based on **S5:** $DTA \triangleleft (ID_{rsu})_{pk'_{dta}}$, **AS 7.** $DTA| \equiv |\xrightarrow{K_{DTA \to RSU}} RSU$, and by utilizing meaning rule, we get:

**S6:** $DTA| \equiv RSU |\sim (RID_{rsu})$.

From **S6:** $DTA| \equiv RSU |\sim (RID_{rsu})$, **AS 2.** $DTA| \equiv \#(T_1, T_2, R_i)$, and by utilizing the rule of freshness and nonce verification, we obtain:

**S7:** $DTA| \equiv RSU| \equiv (RID_{rsu})$

Therefore, the **Goal 3:** $DTA| \equiv RSU| \equiv (RID_{rsu})$ is achieved.

According to **S7:** $DTA| \equiv RSU | \equiv (RID_{rsu})$, **AS 10.** $DTA| \equiv RSU \Rightarrow (RID_{rsu})$ and by utilizing jurisdiction rule, we get: **S8:** $DTA| \equiv (RID_{rsu})$. Thus, the **Goal 4:** $DTA| \equiv (RID_{rsu})$ is accomplished.

According to **SMI 4.** $DTA \to RSU$: $(K_{DTA \to RSU})PK_{rsu}$, we get:

**S9:** $RSU \triangleleft (K_{DTA \to RSU})PK_{rsu}$.

From **S9:** $RSU \triangleleft (K_{DTA \to RSU})(PK_{rsu})$, **AS 12.** $RSU|\equiv |^{K_{DTA} \to RSU} DTA$, and by utilizing message meaning rule, we obtain:

**S10:** $RSU|\equiv DTA| \sim (K_{DTA \to RSU})$.

According to **S10:** $RSU| \equiv DTA| \sim (K_{DTA \to RSU})$, **AS 3.** $RSU| \equiv \#(T_3)$ and by utilizing the freshness rule and nonce verification, we get:

**S11:** $RSU|\equiv DTA| \equiv (K_{DTA \to RSU})$.

Therefore, the **Goal 5:** $RSU| \equiv DTA| \equiv (k_{DTA \to DTA})$ is achieved.

Based on **S11:** $RSU| \equiv DTA| \equiv (K_{DTA \to RSU})$, **AS 13.** $RSU| \equiv DTA \Rightarrow (K_{DTA \to RSU})$ and utilizing the rule of jurisdiction, we obtain:

**S12:** $RSU| \equiv (K_{DTA \to RSU})$.

Thus, the **Goal 6:** $RSU| (k_{dta \to rsu})$ is achieved. From **SMI 5.** $RSU \to V_i : (pk_{MID_v^i})_{h(MID_v^i)}$, we get:

**S13:** $V_i \triangleleft (pk_{(MID_v^i)h(MID_v^i)}$.

According to **S13:** $V_i \triangleleft (pk_{MID_v^i h(MID_v^i)})$, **AS 8.** $V_i | \equiv V_i \xrightarrow{VID} RSU$, and using the rule of the message meaning, we obtain:

**S14:** $V_i | \equiv RSU| \sim (SK_{rsu})$.

From **S14:** $V_i|RSU|(SK_rsu)$, **AS 4.** $V_i | \equiv \#(T_2,R_i)$, and utilizing the freshness rule and nonce-verification, we get:

**S15:** $V_i | \equiv RSU| \equiv (SK_{rsu})$.

Thus, the **Goal 7:** $V_i| \equiv RSU| \equiv (pk'_{dta})$ is achieved.

Based on **S15:** $V_i | \equiv RSU| \equiv (SK_{rsu})$, **AS 11.** $V_i |\equiv RSU \Rightarrow (SK_{rsu})$ and using jurisdiction rule, we obtain:

**S16:** $V_i | \equiv (SK_{rsu})$.

Therefore, the **Goal 8:** $V_i| \equiv pk'_{dta}$ is achieved. Consequently, the proposed scheme's mutual authentication is proven based on achieving the stated goals, and the vehicles can mutually communicate with RSU and DTA.

## THE SIMULATION OF OUR SCHEME USING AVISPA

AVISPA refers to Internet Security Protocols and Applications Automated Validation. It is a web-based push-button tool used to simulate the authentication protocols' security and formally validate them. To code the protocol, AVISPA uses the High-Level Protocol Specification Language (HLPSL). It is made up of four back-ends called HLPSL2IF and a tool for translators. The translator method is used to convert a scheme written in HLPSL to Intermediate Format (IF). This IF is a general language understood by all back-ends and used to evaluate and analyze multiple properties defined in the scheme by different back-ends. Four back-ends are available: Constraint-Logic-based At-tack Searcher (CL-AtSe), On-the-fly Model-Checker (OFMC), Automatic Approximate Tree Automata for Security Scheme Analysis (TA4SP), and SAT-based Model-Checker (SATMC). The architecture of AVISPA is illustrated in Fig. 9 (*Vigano, 2006*; *Chevalier, 2004*). It is crucial to specify designed protocols in the HLPSL language in AVISPA (*Chevalier, 2004*) HLPSL is based on roles: each participant role determines the primary roles, and the scenarios of fundamental roles describe composition roles. Each function is independent of the others and, by requirements, obtains some initial information and then communicates

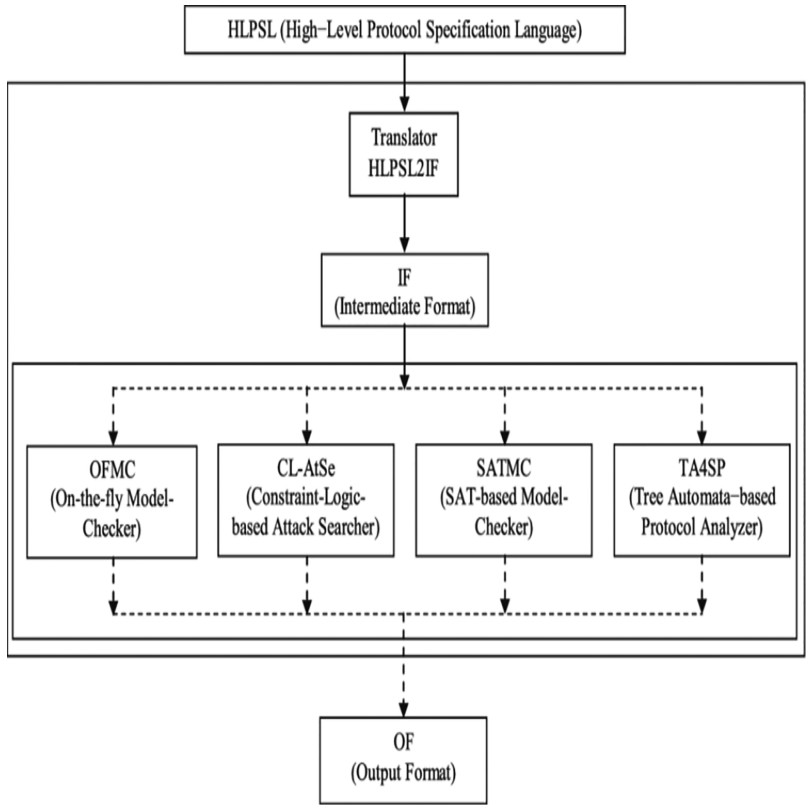

**Figure 9** **The AVISPA structure.**     

with the other roles across channels. The intruder is often modelled in HLPSL using the Dolev-Yao model (*Dolev & Yao, 1983*) (as in the threat model used in this paper) with the possibility of assuming a proper function for the intruder in the running of a protocol. The positioning system decides the number of meetings, the number of principals and the roles. By using one of the four back-ends, the output format (OF) of AVISPA is created. If a protocol analysis (by detecting an attack or not) has been successful, the performance determines precisely what the outcome is and under what conditions it has been obtained. Comprehensive formats for the OF can be found in *Chevalier, 2004*.

## Scheme specification in HLPSL

There are three roles played by the Vi vehicle, RSU road-side unit, and DTA domain trusted authority in the proposed scheme. The other role is the role of the session, environment, and goal. As shown in Figs. 11 and 12, all the specified roles are coded in HLPSL. First, in Figs. 11 and 12, the role played by the vehicle is shown. The agent vehicle Vi receives the start signal $/\backslash$ *RCV* (*start*) = |> and the states changes from 0 to 1. Then, it transmits the registration message (*VIDi.Ri'*. $CT_{vT}A'$. *Ti'_SKvirsu*) to the road-side unite *via* a secure channel $/\backslash$ *SND* () command. The $/\backslash$ *secret*(*VIDi, Ai, Ki, s1, Vi*) declares that the information (*VIDi, Ai, Ki*) is kept secret permanently to the agent Vi, and the label (s1) is the protocol (id) used to identify the goal. The declaration $/\backslash$ *secret*

```
role vehicle (Vi, RSU, DTA : agent, SKvirsu :
symmetric_key,
SND, RCV: channel(dy))
played_by Vi
def=
local State : nat,
VIDi, IDdta, Ki,HIDi : text,
J, K, Q, T,Ti, Ni,Cig, CIDi: text,
TS1, TS2, TS3, TS4, IDrsu, Ri, Rn, Rt, Ii: text,
NIDi, Ai, Bi, SKrsudta, Fi, SKvidta : text,
Gi, Mi,FIDi,X_rsu,Xi : text,
CT_v_TA,Sign_rsu, Sign_vi, CT_v_rsu ,
Ai_dta, CT_v_RSU,CT_RSU_v: text,
H : hash_func, Gen, Rep : hash_func
const vehicle_rsu_ts1, rsu_domainTA_ts2,
domainTA_rsu_ts3, vehicle_rsu_ri, rsu_vehicle_ts4,
domainTA_vehicle_rn,
s1, s2, s3, s4, s5, s6 : protocol_id
init State := 0
transition
%%Vehicle Registration Phase%%%%%%%%%%
1. State = 0 ∧ RCV(start) =|>
State' := 1 ∧ Ti' := H(VIDi.Ki)
∧ Ai':= new()
∧ Ri':= new()
∧ CT_v_TA':= H(Ai'.Ri'.Ti')
∧ SND({VIDi.Ri'.CT_v_TA'.Ti'}_SKvirsu)
∧ secret({VIDi,Ai,Ki},s1,Vi)
∧ secret(VIDi, s2, {Vi,RSU})
∧ secret(SKrsudta, s3, {RSU,DTA})
∧ secret(SKvirsu, s4, {Vi,RSU})
∧ secret({J,K,Q,IDrsu}, s5, RSU)
∧ secret(IDdta, s6, {Vi,RSU,DTA})
%%%%%Joining Phase%%%%%%%%%%%%%%%
2. State = 1 ∧ RCV({{Ai'.VIDi.IDrsu}_J.
xor(H(VIDi.IDrsu.K),
H(VIDi.Ki)). H.Gen.Rep.T}_SKvirsu) =|>
State' := 2 ∧ TS1':= new()
∧ Ri':= new()
∧ Rn':= new()
∧ K':= new()
∧ FIDi':=new()
∧ VIDi':=new()
∧ CT_RSU_v':= new()
∧ Xi':= H(Rn'.K')
∧ Ii':= Xi.H(Mi'.Xi')
∧ Mi':= H(HIDi'.TS1'.Ri')
∧ Ai_dta':= H(Rn.K)
∧ HIDi':= H(VIDi.Rn)
∧ Sign_vi':= ({VIDi'.Ri'.TS1'}.SKvirsu)
∧ CT_v_RSU':= ({Sign_vi'.HIDi'.TS1'.Mi'}.SKvirsu)
∧ CIDi':= {H(VIDi.{Ai'.VIDi.IDrsu}_J.IDdta.Ri'.HIDi'.
TS1') .IDdta.Ri'}_H(VIDi.IDrsu.K)
∧ SND({Ai'.Sign_vi'.CT_v_RSU'.VIDi.IDrsu}_J.CIDi'.
CT_v_RSU'.TS1')
% Vi has freshly generated the values TS1 and r_i for
RSU
∧ witness (Vi,RSU,vehicle_rsu_ts1, TS1')
∧ witness (Vi,RSU,vehicle_rsu_ri, Ri')
% Vi receives the message m4 from RSU
3. State = 2 ∧
RCV({H(VIDi.NIDi'.{FIDi'.VIDi.CT_RSU_v.IDrsu}_J.IDdta.
H(H(NIDi'.IDdta.Ri'.Rn')).Rn'.TS4').
NIDi'.{FIDi'.VIDi.IDrsu}_J.IDdta.xor(Rn',Ri').
H(H(NIDi'.IDdta.Ri'.Rn')).TS4'}_H(VIDi.IDrsu.K).TS4') =|>
State' := 3 ∧ request(RSU, Vi, rsu_vehicle_ts4, TS4')
∧ request(DTA, Vi, domainTA_vehicle_rn, Ri')
end role
```

```
role rsu (Vi, RSU, DTA : agent, SKvirsu : symmetric_key,
SND, RCV: channel(dy))
played_by RSU
def=
local State : nat,
VIDi, IDdta, Ki,FIDi : text,
J, K, Q, T, Ni, Cig, CIDi,MIDi: text,
TS1, TS2, TS3, TS4, IDrsu, Ri, Rn, Rt: text,
NIDi, Ai, Bi, SKrsudta, Fi, SKvidta : text,
Gi, Rg, Rgnew, Cignew, Mi,Xi,Ii,HIDi : text,
CT_v_TA,Sign_rsu, Sign_vi, CT_v_rsu ,
Ai_dta, CT_v_RSU,CT_RSU_v, CT_rsu_dta: text,
H : hash_func, Gen, Rep : hash_func
const vehicle_rsu_ts1, rsu_domainTA_ts2,
domainTA_rsu_ts3,
vehicle_rsu_ri, rsu_vehicle_ts4, domainTA_vehicle_rn,
rsu_dta_ts2, domainTA_rsu_rn,
s1, s2, s3, s4, s5, s6 : protocol_id
init State := 0
transition
1. State = 0 ∧ RCV({VIDi.H(VIDi.Ki)}_SKvirsu)=|>
State' := 1 ∧ secret({IDrsu,IDdta,Ki},s1,Vi)
∧ secret(VIDi, s2, {Vi,RSU}) ∧ secret(SKrsudta, s3,
{RSU,DTA})
∧ secret(SKvirsu, s4, {Vi,RSU}) ∧ secret({J,K,Q,IDrsu}, s5,
RSU)
∧ secret(IDdta, s6, {Vi,RSU,DTA})
∧ Rg' := new() ∧ IDdta':= new()
∧ IDrsu':=new() ∧ TS1':=new()
∧ Ri':= new() ∧ Rn':= new()
∧ K':= new() ∧ Xi':= H(Rn'.K') ∧ Ii':= Xi.H(Mi'.Xi')
∧ Mi':= H(IDrsu'.IDdta'.TS1'.Ri')
∧ Sign_rsu':= ({IDrsu'.IDdta'.TS1'.Ri'}_SKvirsu)
∧ CT_RSU_v':= ({Sign_rsu'. Mi'}_SKvirsu)
∧ Cig' := {Rg'.VIDi.IDrsu}_J
∧ Ni' := xor(H(VIDi.IDrsu.Sign_rsu.K), H(VIDi.Ki.IDdta))
∧ SND({Cig'.Ni'.H.Gen.Rep.T}_SKvirsu)
2. State = 1 ∧ RCV({Rg'.VIDi.IDrsu}_J.
{H(VIDi.{Rg'.VIDi.IDrsu}_J.IDdta.Ri'.TS1')
.IDdta.Ri'}_H(VIDi.IDrsu.
K).TS1')=|> State' := 2 ∧ NIDi' := new()
∧ TS2' := new() ∧ FIDi':= new()
∧ Sign_rsu':= new()
∧ Ai':= xor(Ri', H(SKrsudta.NIDi'.IDdta.TS2'))
∧ Bi' := {H(NIDi'.IDdta.Ri'.TS2').NIDi'.
IDdta.Ai'.TS2'}_SKrsudta
∧ CT_rsu_dta':= ({FIDi'.Sign_rsu'.TS2'}_SKrsudta)
∧ SND(Bi'.TS2')
∧ witness (RSU,DTA,rsu_dta_ts2, TS2')
3. State = 3 ∧ RCV({H(NIDi'.IDdta.Rn'.TS3').
H(SKvidta').NIDi'.IDdta.
xor(Rn', H(SKrsudta.NIDi'.IDdta.
TS3')).TS3'}_SKrsudta.TS3') =|>
State' := 4 ∧ TS4' := new()
∧ Rgnew' := new()
∧ Ri':= new()
∧ MIDi':= new()
∧ IDdta':= new()
∧ Rt' := xor(Rn',Ri)
∧ CT_RSU_v':= (MIDi'.Ri'.TS4')
∧ Mi' := {H(VIDi.NIDi'.IDdta'.IDdta.
H(H(NIDi'.IDdta.Ri.Rn')).Rn'.
TS4'). NIDi'.IDdta'.IDdta.Rt'.
H(H(NIDi'.IDdta.Ri.Rn')).TS4'}_H(VIDi.IDrsu.K)
∧ SND(Mi'.TS4')
∧ witness (RSU,Vi,rsu_vehicle_ts4, TS4')
∧ request(Vi, RSU, vehicle_rsu_ts1, TS1)
∧ request(Vi,RSU,vehicle_rsu_ri, Ri)
∧ request(DTA, RSU, domainTA_rsu_ts3, TS3')
∧ request(DTA,RSU,domainTA_rsu_rn, Rn')
end role
```

**Figure 10  The vehicle and RSU roles in HLPSL.**

```
role domainTA (Vi, RSU, DTA : agent,
SKvirsu : symmetric_key,
SND, RCV: channel(dy))
played_by DTA
def=
local State : nat,
VIDi, IDdta, Ki, MIDi : text,
J, K, Q, T, Ni, Cig, CIDi: text,
TS1, TS2, TS3, TS4, IDrsu, Ri, Rn,Xi, Rt: text,
NIDi, Ai, Bi, SKrsudta, Fi, SKvidta, SKi : text,
Gi, Mi, SKrsuvi, SKmidi, CT_DTA_vi : text,
H : hash_func, Gen, Rep : hash_func
const vehicle_rsu_ts1, rsu_domainTA_ts2, domainTA_rsu_ts3,
vehicle_rsu_ri, rsu_vehicle_ts4, domainTA_vehicle_rn,
domainTA_rsu_rn, rsu_domainTA_ri,
s1, s2, s3, s4, s5, s6 : protocol_id
init State := 0
transition
% Authentication and key agreement phase
% DTA receives authentication request m2 from RSU
1. State = 0 ∧ RCV({H(NIDi'.IDdta.Ri'.TS2').NIDi'.
IDdta.xor(Ri', H(SKrsudta.NIDi'. IDdta.TS2')).TS2'}_SKrsudta.TS2')=|>
State' := 1  ∧ secret({IDrsu,IDdta,Ki},s1,Vi)
        ∧ secret(VIDi, s2, {Vi,RSU})
        ∧ secret(SKrsudta, s3, {RSU,DTA})
        ∧ secret(SKvirsu, s4, {Vi,RSU})
        ∧ secret({J,K,Q,IDrsu}, s5, RSU)
        ∧ secret(IDdta, s6, {Vi,RSU,DTA})
        ∧ Rn' := new()
        ∧ K':= new()
        ∧ MIDi':= new()
        ∧ SKi':= new()
        ∧ Xi':= H(Rn'.K')
        ∧ TS3' := new()
        ∧ SKrsuvi':= (Rn'.Xi')
        ∧ SKmidi':= (Rn'.SKi')
        ∧ Fi' := xor(Rn', H(SKrsudta.NIDi'.IDdta.TS3'))
        ∧ SKvidta' := H(NIDi'.IDdta.Ri'.Rn')
        ∧ Gi' := {H(NIDi'.IDdta.Rn'.TS3'). H(SKvidta').NIDi'.IDdta.Fi'.
TS3'}_SKrsudta
        ∧ CT_DTA_vi':= ({MIDi'.SKmidi'.Ri'}_SKvirsu)
        ∧ SND(Gi'.CT_DTA_vi'.TS3')
        ∧ witness (DTA,RSU,domainTA_rsu_ts3, TS3')
        ∧ witness (DTA,RSU,domainTA_rsu_rn, Rn')
        ∧ request(RSU, DTA, rsu_domainTA_ts2, TS2')
        ∧ request(RSU, DTA, rsu_domainTA_ri, Ri')
end role
```

**Figure 11  The DTA role in HLPSL.** 

($SKrsudta$, $s3$, $RSU$, $DTA$) indicates that the value ($SKrsudta$) is shared between the RSU and DTA using the label ($s3$). While, the declaration $/\backslash$ *secret* ($SKvirsu$, $s4$, $Vi$, $RSU$) shows that the value ($SKvirsu$) is known to the Vi and RSU. The identity of the domain trusted authority ($IDdta$) used in the declaration $/\backslash$ *secret* ($IDdta$, $s6$, $Vi$, $RSU$, $DTA$)and stated that it is known to the agents' Vi, RSU, and DTA. In the login phase, the vehicle sends the message $/\backslash$ *SND* ($Ai'$. $Sign\_vi'.CT\_v\_RSU'.VIDi. IDrsu\_J.CIDi'.CT\_v\_RSU'.$ $TS1'$) using $/\backslash$ *SND* () command, and the declarations $/\backslash$ witness (Vi,RSU,vehicle_rsu_ts1, TS1'), and $/\backslash$*witness*($Vi$, $RSU$, $vehicle\_rsu\_ri$, $Ri'$) indicates that the timestamp (TS1), and (Ri) have generated freshly by the vehicle for the RSU. State 3 shows that the vehicle receives $/\backslash$ *RCV* ($H$ ($VIDi.NIDi'$. $FIDi'$. VIDi CT_RSU_v. IDrsu_J. IDdta. H(H(NIDi'.

```
role session(Vi, RSU, DTA: agent,
SKvirsu : symmetric_key)
def=
local US, UR, SS, SR, VS, VR: channel (dy)
composition
vehicle(Vi, RSU, DTA, SKvirsu, US, UR)
/\ rsu(Vi, rsu, DTA, SKvirsu, SS, SR)
/\ domainTA(Vi, rsu, DTA, SKvirsu, VS, VR)
end role
%%%%%%%%%%%%%%
role environment()
def=
const vi, rsu, dta : agent,
skvirsu : symmetric_key,
h : hash_func,
gen, rep : hash_func,
ts1, ts2, ts3, ts4 : text,
vehicle_rsu_ts1, rsu_domainTA_ts2,
domainTA_rsu_ts3, vehicle_rsu_ri,
rsu_vehicle_ts4, domainTA_vehicle_rn,
domainTA_rsu_rn, rsu_domainTA_ri,
s1, s2, s3, s4, s5, s6 : protocol_id
intruder_knowledge = {h, gen, rep, ts1, ts2, ts3, ts4}
composition
session(vi, rsu, dta, skvirsu)
/\ session(vi, rsu, dta, skvirsu)
/\ session(vi, i, dta, skvirsu)
/\ session(vi, rsu, i, skvirsu)
end role goal
secrecy_of s1
secrecy_of s2
secrecy_of s3
secrecy_of s4
secrecy_of s5
secrecy_of s6
authentication_on vehicle_rsu_ts1, vehicle_rsu_ri
authentication_on rsu_domainTA_ts2, rsu_domainTA_ri
authentication_on domainTA_rsu_ts3, domainTA_rsu_rn
authentication_on rsu_vehicle_ts4, rsu_dta_ts2
authentication_on domainTA_vehicle_rn
end goal
environment()
```

**Figure 12 Role specification of the proposed scheme in HLPSL for the session, goal, and environment.**

IDdta.Ri'.Rn')).Rn'.TS4'), and the declarations $/\backslash$ *request*($RSU$, $Vi$, $rsu\_vehicle\_ts4$, $TS4'$), and $/\backslash$ *request*($DTA$, $Vi$, $domainTA\_vehicle\_rn$, $Ri'$) indicates the vehicle acceptance of the timestamp that generated by the RSU, and the (Ri) that sent by the DTA. The role specification of the role played by the RSU is shown in Fig. 10B. The RSU computes the necessary parameters after receiving the message ($VIDi.H(VIDi.Ki)_SKvirsu$) through a secure channel.

The declaration *secret* (*IDrsu*, *IDdta*, *Ki*, *s*1, *Vi*) indicates that the values are kept secret to the Vi using the label (s1). The secret (*VIDi*, *s*2, *Vi*, *RSU*) declaration shows that VIDi is shared between the Vi and the RSU. The statement secret (*SKrsudta*, *s*3, *RSU*, *DTA*) states that SKrsudta is shared between RSU and DTA. At the same time, *secret* (*SKvirsu*, *s*4, *Vi*, *RSU*) indicates SKvirsu is known to the Vi and RSU. In the authentication phase, the RSU sends the message (Mi'.TS4') *via* a secure channel using SND ( ). However, the witness (*RSU*, *Vi*, *rsu_vehicle_ts*4, *TS*4′) declaration specifies that the RSU has freshly generated TS4 for the vehicle. The declaration request (*Vi*, *RSU*, *vehicle_rsu_ri*, *Ri*) indicates that the vehicle accepts Ri's value. The specification of domain trusted authority role (domainTA) is shown in Fig. 11. The DTA receives the message ({*H* (*NIDi*′. *IDdta.Ri*′. *TS*2′). *NIDi*′. *IDdta.xor* (*Ri*′, *H* (*SKrsudta. NIDi*′. *IDdta.*)).*TS*2′} *SKrsudta*) from the RSU. However, the declaration secret (*SKrsudta*, *s*3, *RSU*, *DTA*) indicates that the value SKrsudta is shared between the RSU and DTA using the label (s3: protocol_id). In the command *secret* (*SKvirsu*, *s*4, *Vi*, *RSU*), we declare that the SKvirsu shared between the vehicle and RSU generated freshly by the DTA. The value IDdta as stated in declaration *secret* (*IDdta*, *s*6, *Vi*, *RSU*, *DTA*) is known to the vehicle, RSU, and DTA. Later, the domain trusted authority sends the message (*Gi*′. *CT_DTA_v i*′.*TS*3′) using secure channel SND (). Nevertheless, the declarations witness (*DTA*, *RSU*, *domainTA_rsu_ts*3, *TS*3′, and *witness*(*DTA*, *RSU*, *domainTA_rsu*, *Rn*′) states that the DTA has freshly generated TS3', and Rn' for the RSU. We presented the roles for the session, goal, and environment in the HLPSL code in Fig. 12. All primary roles, including roles for the (Vi, RSU, and DTA), are incorporated with concrete arguments in the session segments. The environment section contains the global constant and composition of one or more sessions, and knowledge of the intruder is also provided. We define six secrecy objectives in our scheme simulation, and five authentications are tested.

- The secrecy_of s1: It represents that the (VIDi, Ai, Ki) is kept secret only (Vi).
- The secrecy_of s2: It states that the (VIDi) is known secretly (Vi, RSU).
- The secrecy_of s3: It indicates that the value (SKrsudta) is shared secretly (RSU, DTA).
- The secrecy_of s4: The (SKvirsu) is secretly shared between the Vi and RSU.
- The secrecy_of s5: indicates that the (J, K, Q, IDrsu) is known (RSU).
- The secrecy_of s6: It states that the identity (IDdta) is known to all entities (Vi, RSU, DTA).
- The authentication_on vehicle_rsu_ts1, vehicle_rsu_ri: It represents that the values (TS1′), and (Ri′) are generated randomly and known to the (Vi) and (RSU).
- The authentication_on rsu_domainTA_ts2, rsu_domainTA_ri: It indicates that the values (TS3′), and (Rn′) are generated by the DTA and sent to the RSU securely, and the values are fresh.
- The authentication_on domainTA_rsu_ts3, domainTA_rsu_rn: The values TS3′ and Rn ′ are generated freshly for the RSU by the DTA and authenticates the RSU to DTA.

```
% OFMC
% Version of 2006/02/13
SUMMARY
 SAFE
DETAILS
 BOUNDED_NUMBER_OF_SESSIONS
PROTOCOL
 /home/span/span/testsuite/results/ProposedScheme.if
GOAL
 as_specified
BACKEND
 OFMC
COMMENTS
STATISTICS
 parseTime: 0.00s
 searchTime: 0.12s
 visitedNodes: 16 nodes
 depth: 4 plies
```

(a) The OFMC result.

```
SUMMARY
 SAFE

DETAILS
 BOUNDED_NUMBER_OF_SESSIONS
 TYPED_MODEL

PROTOCOL
 /home/span/span/testsuite/results/ProposedScheme.if

GOAL
 As Specified

BACKEND
 CL-AtSe

STATISTICS

 Analysed  : 3 states
 Reachable : 0 states
 Translation: 0.11 seconds
 Computation: 0.00 seconds
```

(b) CL-AtSe results.

**Figure 13 (A0B) The simulation results of the proposed scheme.**

- The authentication_on rsu_vehicle_ts4, rsu_dta_ts2: It represents that the timestamp TS2′ is generated freshly by the RSU for the vehicle.
- The authentication_on domainTA_vehicle_rn: indicates that the value Rn′ generated freshly by the DTA for the vehicle.

## Simulation results

For an execution test and a limited number of model checking sessions, we chose the back end OFMC (*Basin, Mödersheim & Vigano, 2005*). This back-end tests whether legitimate agents can execute the specified protocol by conducting a passive intruder search for replay attack checks. After that, the intruder is given the information of some regular sessions between the legitimate agents by this back-end. This back end also checks whether the attacker can carry out any man-in-the-middle attack for the Dolev-Yao model search. With the OFMC back-end, under the AVISPA web tool, we simulated our schema for formal security verification. Figures 13A and 13B in Fig. 13 show the simulation results for our scheme's formal security verification using OFMC. The first written part, called the Summary, indicates in these statistics whether the protocol is stable, risky, or whether the analysis is inconclusive. The written Overview segment safeguards our scheme. The information section explains what state the device is considered secure, what conditions were used to detect an attack, or why the analysis was inconclusive.

 

**Table 5 The execution time of different cryptographic operations.**

| Cryptographic operation | Time (ms) |
| --- | --- |
| Bilinear pairing operation $(T_{BP})$ | 4.211 |
| Scalar multiplication bilinear pairing in $G_1$ $T_{sm-bp}$. | 1.5654 |
| Point addition of the bilinear pairing in $G_1$ $T_{pa-pb}$. | 0.0106 |
| Map- to-point of the bilinear pairing in $G_1$ $T_{mtp}$. | 4.1724 |
| Scalar multiplication of the ECC $T_{sm-ecc}$. | 0.6718 |
| Point addition of the ECC in an additive group G $T_{pa-ecc}$. | 0.0031 |
| Hash function $T_h$ | 0.001 |
| Point exponentiation $T_{pe}$ | 9.0082 |
| Symmetrical encryption $(T_{se})$ | 0.0046 |
| Symmetrical decryption $(T_{sd})$ | 0.0046 |
| Asymmetric signature $(T_{as})$ | 3.8500 |
| Asymmetric signature verification $(T_{av})$ | 0.1925 |

It is recognized that our architecture is deemed to be protected, and our system does not detect an attack. Consequently, the result of this figure suggests that our system is safe from passive and active attacks, including man-in-the-middle replay attacks and attacks. Knowledge of daily sessions between the authentic agents is given to the intruder. Figures 13A and 13B in Fig. 13 show the OFMC and CL-AtSe back-end simulation results and demonstrate that the scheme is secure and stable against attacks.

# PERFORMANCE EVALUATION

In this section, we evaluate the performance of the proposed system in terms of cost of computation and communication with other VANET authentication schemes, *e.g.*, ID-CPPA (*Ali & Li, 2020*); AAAS (*Jiang, Ge & Shen, 2020*), and HCDA (*Tan, Xuan & Chung, 2020*). The performance of the schemes against those schemes is shown in Table 6. The performance metrics evaluation is described as following:

## Computation cost

Here, we analyze the computation cost of the proposed scheme against other authentication schemes for the VANET system, *e.g.*, ID-CPPA (*Ali & Li, 2020*); AAAS (*Jiang, Ge & Shen, 2020*), and HCDA (*Tan, Xuan & Chung, 2020*) are summarized in Table 6. In this study, the cryptographic operations involved are counted. To represent the comparison, Table 5 shows the notations, definition, and calculation of their estimated execution time by using the PBC library stated by *Al-Shareeda et al. (2020)* for different cryptographic operations. It is noted that the XOR operation and concatenated operation k are ignored because their execution time is negligible. The proposed scheme's simulation was carried out on Intel Core™i7-5700HQ, CPU 2.70 GHz platform using Java Pairing-Based Cryptography Library (JPBC) library. In the proposed scheme, we applied five cryptographic operations hash function, symmetrical encryption, symmetrical decryption, asymmetric signature, and asymmetric signature verification that related to AES and RSA algorithm, which are respectively donated as $T_h$, $T_{se}$, $T_{sd}$, $T_{as}$, and $T_{av}$.

**Table 6 Comparison of the computation and communication costs of schemes.**

| Scheme | Computation cost (ms) | | | | Communication cost (bits) |
|---|---|---|---|---|---|
| | Vehicle side ($V_i$) | RSU side | TA side | Total | |
| ID-CPPA (*Ali & Li, 2020*) | $3T_{BP} \approx 12.633\ ms$ | $T_{sm-bp} + T_{BP}$ $\approx 5.776\ ms$ | $1T_{sm-bp}$ $+ 2T_{BP} \approx 9.9874\ ms$ | 28.3964 ms | 2,432 bits |
| AAAS (*Jiang, Ge & Shen (2020)*) | $2T_{sm-bp} + 1T_{BP}$ $\approx 7.3418\ ms$ | $1T_{sm-bp} + 1T_{BP} +$ $1T_{mtp} \approx 9.9488\ ms$ | $3T_{sm-bp} + 1T_{BP}$ $+ 1T_{mtp} \approx 13.0796\ ms$ | 30.3702 ms | 3,264 bits |
| HCDA (*Tan, Xuan & Chung, 2020*) | $2T_h + 1T_{pe} + 1T_{sm-bp}$ $\approx 10.5756\ ms$ | $2T_h + 2T_{pe}$ $\approx 18.0184\ ms$ | $2T_h \approx 0.002\ ms$ | 28.596 ms | 2,528 bits |
| Proposed scheme | $3T_h + 1T_{as} + 1T_{se} + 1T_{sd}$ $+ 1T_{av} \approx 4.0547\ ms$ | $1T_h + 1T_{as} + 2T_{se}$ $+ 2T_{sd} + 1T_{av} \approx 4.0619\ ms$ | $1T_{se} + 1T_{sd} \approx 0.0092\ ms$ | 8.1258 ms | 1,408 bits |

The utilized operations execution time is independently 0.001 ms, 0.0046 ms, 0.0046 ms, 3.8500 ms, and 0.1925.

In ID-CPPA Scheme *Ali & Li (2020)*, the vehicle needs to execute three times bilinear pairing operation $3T_BP$ that has the execution time 4.211 ms, and it related to the ECC algorithm, thus, the computation cost in the vehicles side was $3T_BP \approx 12.633\ ms$. In the RSU side, there were two cryptographic operations Scalar multiplication bilinear pairing in G1 $T_{sm-bp}$, and bilinear paring operation $T_{BP}$. The $T_{sm-bp}$, and $T_{BP}$ have been used one time only for each. Thus, the computation cost is $T_{sm-bp} + T_{BP} \approx 5.776\ ms$. In the trusted authority side, it needs to execute $1T_{sm-bp}$, and $2T_{BP}$, and their execution time is $\approx 9.9874\ ms$. Therefore, the total computation cost of Ali's scheme (*Ali & Li, 2020*) is approximately $\approx 28.3964\ ms$. In AAAS (scheme *Jiang, Ge & Shen, 2020*), the message $< f_v^i, Exp_{f_v^i}, TS_4, N_8 >$ is signed by the vehicle for authentication, and computes the signature $\alpha = V_v, W_v$, where $V_v = r_v p$, $W_v = r_v^{-1} sk_i + H_2(f_v^i \| Exp_{f_v^i} \| TS_4 \| N_8, V_v) b_i$, and select a random number $r_v \in Z_q^*$. Later, it sends $< f_v^i, Exp_{f_v^i}, TS_4, N_8, \alpha >$ to the RSU. After the RSU receives he message, it checks $e(f_v^i, P_{pub}, f_v^i) e(V_v, H_2((f_v^i \| Exp_{f_v^i} \| TS_4 \| N_8, V_v)) == e(V_v, f_v^i W_v)$ to verify the signature. The scheme performed six-point multiplication operations $6T_{sm-bp}$, three bilinear map operations $3T_{BP}$, and two map-to-point hash function $2T_{mtp}$. operation in G1. Therefore, the total computation cost of Jiang scheme (*Jiang, Ge & Shen, 2020*) is equal to $\approx 30.3702\ ms$.

In the HCDA scheme (*Tan, Xuan & Chung, 2020*), it applied three cryptographic operations hash function, point exponentiation, scalar multiplication bilinear pairing in G1, and they are respectively donated as $T_h$, $T_{pe}$, and $T_{sm-bp}$. The estimated execution time is 0.001, 9.0082, and 1.5654 independently. However, the vehicle needs to apply two times hash function $2T_h$, one-time exponentiation operation $1T_{pe}$, and multiplication operation $1T_{sm-bp}$, thus, the computation cost in vehicle side is $\approx 10.5756\ ms$. In RSU side,

two-time hash function $2T_h$, and two-times exponentiation operation $2T_{pe}$, and the computation cost in RSU is nearly ≈ 18.0184 $ms$. In the TA side, there were two times hash function operation used $2T_h$ and it costs 0.002 ms. Therefore, the total computation cost of Tan's scheme (*Tan, Xuan & Chung, 2020*) is approximately ≈ 28.596 $ms$. In the proposed scheme, the vehicle needs to execute three times hash function $3T_h$, one times asymmetric encryption $1T_{as}$, one times symmetric encryption $1T_{se}$, one times symmetric decryption $1T_{sd}$ , and one times asymmetric signature verification $1T_{av}$ related to RSA, and AES. The execution time of these operation is approximately 0.003, 3.8500, 0.0046, 0.0046, and 0.1925 respectively. Therefore, the computation cost in the vehicle side is $3T_h +$ $1T_{as} + 1T_{se} + 1T_{sd} + 1T_{av}$ ≈ 4.0547 $ms$. In the RSU side, there are five operations needed to be executed $e.g.$, one-time hash function $1T_h$, one-time asymmetric encryption $1T_{as}$, two times symmetric encryption $2T_{se}$, two times symmetric decryption $2T_sd$, and one-time asymmetric signature verification $1T_{av}$. Their execution time is independently 0.001, 3.8500, 0.0092, 0.0092, and 0.5775 ms. Therefore, the computation cost in RSU side is $1T_h$ $+ 1T_{as} + 2T_{se} + 2T_{sd} + 1T_{av}$ ≈ 4.0619 $ms$. Likewise, the DTA needs to execute two cryptographic operations, one-time symmetric encryption $1T_{se}$, and one time symmetric decryption $1T_{sd}$, The execution time of these operations is 0.0046 ms, and 0.0046 ms. Thus, the computation cost in the DTA side is $1T_{se} + 1T_{sd}$ ≈ 0.0092 $ms$. Therefore, the total computation cost of the proposed scheme is approximately 8.1258 ms. Comparing to other schemes and as shown Table 6, the proposed scheme has less computation cost due to the use of lightweight cryptographic operations which makes the scheme suitable for Industrial IoT environment.

## Communication cost

The communication cost refers to the size of the interacted messages between the system entities. Our proposed scheme has four interacted messages exchanged in the whole joining phase amongst the vehicle, road-side units, and domain trusted authority. 32 bits represent the size of the identity, general hash function 160 bits, secret value 160 bits, time expiration of the value, and the timestamp with the size of 32 bits, respectively. In AAAS scheme (*Jiang, Ge & Shen, 2020*), the message $\alpha = V_v, W_v, V_v, W_v \in$ $G_1, N_8 \in Z_q^*$ with pseudo-identity $f_v^i$ , expiration $Exp_{f_v^i}$, timestamp $TS_4$, and challenge value $N_8$ is signed by the vehicle and transmitted to the RSU. As we mentioned above, the size of the identity is represented as 32 bits, expiration and time stamp is represented as 32 bits, and the challenge value is represented as 1,024 bits. The communication can be calculated as 160 + 32 + 32 + 16 + 1024 × 2. Therefore, the total communication cost of In Jiang scheme (*Jiang, Ge & Shen, 2020*), is 2,432 bits. In ID-CPPA Scheme *Ali & Li (2020)*, the vehicle needs to transmit the message $\alpha_i = (A_i, B_i ) \in G_1$ along together with the pseudo-identity $PIDi = (PID_i, 1, PID_i, 2)$, $where PID_i, 1 \in G_1$, and $PID_i, 1, 2 \in Z_q^*$. However, in their scheme, they take the signature's size in the message and the corresponding identity only into account. Thus, the communication cost of Ali's Scheme (*Ali & Li, 2020*) can be calculated as 128 ÷ 3 + 20 + 4 = 408 bytes, where, (128 bytes = 1,024 bits), (20 bytes = 160 bits), and (4 bytes = 32 bits), therefore, the total communication cost of their

scheme is 3264 bits. In the HCDA scheme *Tan, Xuan & Chung (2020)*, the vehicle publishes a set of parameters $< Request, TS_3^j, ID_j, \phi_j, Cert_v^j >$ with the RSU for mutual authentication. The vehicle is generates requesting packet $< TS_4^i, ID_j^1, Cert_{RSU}^j, \phi_j >$ and sent to the RSU. Hence, the communication cost in the vehicle side is $32 \times 13 + 256 \times 3 + 160 \times 2 + 24 \times 3 = 1,576$ bits. In the RSU, uses an acknowledgment packets $< TS_2^i, ID_{RSU}^i, O_i, hbar_i, R^i, Cert_{RSU}^i >$ and the communication cost can be calculated as $32 \times 6 + 256 \times 1 + 160 \times 3 + 24 \times 1 = 952$ bits. Therefore, the total communication cost of Tan's scheme (*Tan, Xuan & Chung, 2020*) is 2,528 bits. The vehicle sends the message in the proposed scheme $CT_{v \to rsu_1/DTA} = Enc\_V_{v \to rsu_1/DTA}^{aes}\{Sign_v \| FID_v \| T_2 \| M_{rsu}'\}$, where the $Sign_v = Sign\_sk_v\{FID_v, t_{exp}^v, T_2, R_i\}$, The size of the message can calculated as $256 + 32 + 32 + 160 = 480$ bits. Also, the RSU sends the message $CT_{rsu_1 \to v} = Enc\_V_{v \to rsu_1}^{aes}\{t_{exp}^v \| FID_v \| CT_{v \to DTA}\}$ to the DTA , where is $CT_{v \to DTA} = Enc\_CTv \to RSU_1\{R_i\}$ needs $32 + 32 + 160 = 224$ bits. In the DTA side, it needs to send the message $CT_{DTA \to v} = Enc\_K_{DTA \to v}\{MID_v^i, sk_{MID_v^i}', t_{exp}^{MID_v^i}, R_i\}$ to the RSU and needs $32 + 128 + 32 + 160 = 352$ bits. Later, the RSU will perform the same length of the message to forward it to the vehicle which costs 352 bits. Therefore, if the proposed system is 1408 bits, the total communication cost. Therefore, the comparison of the cost of communication as shown in Table 6 indicates that the proposed system has a lower cost of communication relative to other systems.

# CONCLUSION

This paper presents a lightweight online and offline cross-domain authentication scheme to support the large-scale industrial IoT environment of the VANET system. The scheme aimed to support the domain vehicles and reduce the system workload by adding a domain trusted authority. To support offline authentication, the scheme enables the automotive industrial to preload the secret credentials and information into the vehicles in their prior deployment to enable them to authenticate wherever the network's connectivity is unavailable. The study proposed a lightweight cryptographic method by combining asymmetric and symmetric cryptographic algorithms AES and RSA to ensure confidentiality, authentication, and data integrity. This combination performs a lightweight cryptographic operation and takes advantage of the AES-RSA algorithm since they require less computation. The security of the VANET system is improved due to the secure transmission and verification process, making it secure against such known attacks replay attack, modification attack, impersonation attack, and brute-force attacks. The system's security is checked using the well-known AVISPA security verification tool. Also, using BAN logic, mutual authentication of the scheme is verified. The results indicate that by testing it informally, our scheme achieves some security requirements and attacks. It also showed that the scheme provides better efficiency in terms of communication and cost of computation. In the future, we plan to implement the proposed scheme in the automotive industry for complete offline authentication functionality.

### Funding

The University Putra Malaysia supported this work as part of the "Matching Grant UPM-Kyutech" and Ajman University provided an "Ajman University Graduate Student Grant". The funders had no role in study design, data collection and analysis, decision to publish, or preparation of the manuscript.

### Grant Disclosures

The following grant information was disclosed by the authors:
University Putra Malaysia.
Ajman University.

### Competing Interests

The authors declare that they have no competing interests.

### Author Contributions

- Haqi Khalid conceived and designed the experiments, performed the experiments, analyzed the data, performed the computation work, prepared figures and/or tables, and approved the final draft.
- Shaiful Jahari Hashim conceived and designed the experiments, analyzed the data, prepared figures and/or tables, authored or reviewed drafts of the paper, and approved the final draft.
- Sharifah Mumtazah Syed Ahmad conceived and designed the experiments, authored or reviewed drafts of the paper, and approved the final draft.
- Fazirulhisyam Hashim conceived and designed the experiments, authored or reviewed drafts of the paper, review, and approved the final draft.
- Muhammad Akmal Chaudhary conceived and designed the experiments, authored or reviewed drafts of the paper, review, and approved the final draft.

### Data Availability

The simulation code and the security validation code are available in the Supplemental Files.

### Supplemental Information

Supplemental information for this article can be found online at http://dx.doi.org/10.7717/peerj-cs.714#supplemental-information.

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
