# Peer review of "A lightweight and secure online/offline cross-domain authentication scheme for VANET systems in Industrial IoT"

_PeerJ Computer Science, doi:10.7717/peerj-cs.714_

## Round 0.1 · original submission · Minor Revisions

Dear Authors,

Based on the comments received from the reviewers and my own observation, I recommend minor revisions for the article.

Reviewer 1 ·

Basic reporting

- Please improve the overall readability of the paper.
- The objectives of this paper need to be polished. Contribution list should be polished at the end of the introduction section and last paragraph of the introduction should be the organization of the paper.
- In the first four paragraphs of literature review section, the authors have presented a good references, but they need to present the recent and most updated references.
- Some Paragraphs in the paper can be merged and some long paragraphs can be split into two. (i.e., Related Work first paragprah of more than 2pages)
- Remomve the authors name from table 1 first column. Just give reference number.

Experimental design

- Please cite each equation and clearly explain its terms.
- Clearly highlight the terms used in the algorithm and explain them in the text.
- The procedures and analysis of the data are seen to be unclear.

Validity of the findings

- What are the evaluations used for the verification of results?
- Provide a comparison with existing studies.

Additional comments

- Make sure the Conclusion succinctly summarizes the paper. It should not repeat phrases from the Introduction!
- Authors should add the most recent reference:
1) CANintelliIDS: Detecting In-Vehicle Intrusion Attacks on a Controller Area Network using CNN and Attention-based GRU, IEEE Transactions on Network Science and Engineering
2) Anomaly Detection in Automated Vehicles Using Multistage Attention-Based Convolutional Neural Network, IEEE Transactions on Intelligent Transportation Systems

lastly, Proofread the paper

Reviewer 2 ·

Basic reporting

- Reference format must be standard in the whole paper.

Experimental design

- It is recommended that more information adds for simulation environments, parameters and data.

Validity of the findings

It is ok.

Additional comments

- "VANET security section" needs more detail and references for challenges and criteria identification.
- Response to reviewer must be in an individual file.

Reviewer 3 ·

Basic reporting

propose an online/offline lightweight authentication scheme for VANET cross-domain system in IIoT. The proposed scheme utilizes an efficient AES-RSA algorithm to achieve integrity and confidentiality of the message. The offline joining is added to avoid remote network intrusions and the risk of network service interruptions. The Burrows Abdi Needham (BAN logic) logic is used to prove that this scheme is mutually authenticated. The system’s security has been tested using the well-known AVISPA tool to formally evaluate and verify its security.

Experimental design

propose an online/offline lightweight authentication scheme for VANET cross-domain system in IIoT. The proposed scheme utilizes an efficient AES-RSA algorithm to achieve integrity and confidentiality of the message. The offline joining is added to avoid remote network intrusions and the risk of network service interruptions. The Burrows Abdi Needham (BAN logic) logic is used to prove that this scheme is mutually authenticated. The system’s security has been tested using the well-known AVISPA tool to formally evaluate and verify its security.

Validity of the findings

propose an online/offline lightweight authentication scheme for VANET cross-domain system in IIoT. The proposed scheme utilizes an efficient AES-RSA algorithm to achieve integrity and confidentiality of the message. The offline joining is added to avoid remote network intrusions and the risk of network service interruptions. The Burrows Abdi Needham (BAN logic) logic is used to prove that this scheme is mutually authenticated. The system’s security has been tested using the well-known AVISPA tool to formally evaluate and verify its security.

Additional comments

• The authors should emphasize the difference between other methods to clarify the position of this work further.
• The Wide ranges of applications need to be addressed in Introductions
• The objective of the research should be clearly defined in the last paragraph of the introduction section.
• Add the advantages of the proposed system in one quoted line for justifying the proposed approach in the Introduction section.
• The motivation for the present research would be clearer, by providing a more direct link between the importance of choosing your own method.
Under blockchain background, the following papers can be referred
A survey on blockchain for big data: Approaches, opportunities, and future directions
Toward Blockchain for Edge-of-Things: A New Paradigm, Opportunities, and Future Directions

·

Basic reporting

1.Suggestion: Figure 1- The typical architecture of VANETs in page 2; TA has registered OBUs and RSUs. This can be added to the picture.

2. Suggestion: Page 3; Line#105,106 In the offline authentication, TA is not involved in the joining procedure since the information is preloaded prior.

The message is not coming out clearly what is been preloaded in advance. A pictorial representation can help

3. Correction: Page 9, Figure 3 - The AES-RSA algorithm work diagram; The final outcome after the comparison of private key and public key comparison is Data is secure not Date is secure

4. Correction: Page 13, The DAT then concatenated the signature with the message CTDTA−→RSU = Enc DTAaes {Signdta ∥ M′ }; The DTA (Domain Trust Authority) not DAT

5. Grammar: Page 16, the movement from RSU1 to RSU2 is occur dynamically; can be changed to does occur or is occurring

Experimental design

Question: Page 26; Line 726 - 730 does every RSU preload with vehicle credentials (rvj,SKvj,texp,Ri,TIDv) . What if there are 1000s of RSUs and 1000s of OBU crossing every day?

Recommendation: Please quote an example for man in the middle attack or traceability is a concern with an existing VANETs

Validity of the findings

no comment

---

## Round 0.2 · Minor Revisions

Dear Authors,

I was asked by the editorial board to take over the handling of this manuscript.

The authors have done a great job in improving by addressing most of the comments. However, there are some minor corrections required as per reviewer2. Hence I recommend minor revision for the paper.

Reviewer 1 ·

Basic reporting

-

Experimental design

-

Validity of the findings

-

Additional comments

Accept

Reviewer 2 ·

Basic reporting

- It is recommended to check the formats of figures, tables, and references in the journal's standard format for more readability.
- It is recommended that short sentences be written to make the article more readable.
- In the abstract and introduction, attributes are used relatively, for example, "more security" or "less costly." To better understand, it is necessary to either mention the studied parameters or compare them numerically and as a percentage.
- In the previous works section, it is necessary to check the table with explanations. Some papers are not mentioned in the table. For example, in line 178, a source from 2021 is given, which is not mentioned in the table.

Experimental design

no comment

Validity of the findings

- For equations used in section 6.1, no reference is mentioned, and on the other hand, they are used to compare methods. Therefore, if the authors extract the equations, they add other standard parameters for evaluations.

Additional comments

Given that there is no reference to equations, it seems that the evaluation part needs to be improved

Reviewer 3 ·

Basic reporting

The authors have addressed all of my comments, paper can be accepted in the current form. Thank you for giving me this opportunity.

Experimental design

Good

Validity of the findings

Good

Additional comments

The authors have addressed all of my comments, paper can be accepted in the current form. Thank you for giving me this opportunity.

·

Basic reporting

Suggested contents have been modified.
Looks good

Experimental design

Suggested contents have been modified.
Looks good

Validity of the findings

data looks good.

---

## Round 0.3 · accepted · Accept

Date: 24/08/2021

Authors have addressed all the comments from the reviewers. Hence, this manuscript is recommended for publication in this journal in its current form. Well done for this round of revision.

Reviewer 2 ·

Basic reporting

No comment

Experimental design

No comment

Validity of the findings

No comment

Additional comments

Authors have addressed the reviewer’s comments satisfactorily.